# Measurement Report: Biogenic VOC emission profiles of rapeseed leaf litter and its SOA formation potential

Letizia Abis[1] *, Carmen Kalalian[1], Bastien Lunardelli[1], Tao Wang[2], Liwu Zhang[2], Jianmin Chen[2], Sébastien Perrier[1], Benjamin Loubet[3], Raluca Ciuraru [3] and Christian George[1]

[1] Univ Lyon, Université Claude Bernard Lyon 1, CNRS, IRCELYON, F-69626, Villeurbanne, France.
[2]Shanghai Key Laboratory of Atmospheric Particle Pollution and Prevention, Department of Environmental Science & Engineering, Fudan University, Shanghai,200433, Peoples' Republic of China.
[3]INRAE, UMR ECOSYS, AgroParisTech, Université Paris-Saclay, 78850, Thiverval-Grignon, France
*Now working at: Technische Universität Berlin, Umweltchemie und Luftrinhaltunz, Straße des 17. Juni 135, Berlin, 10623, Germany.

*Correspondence to*: Christian George (christian.george@ircelyon.univ-lyon1.fr)

**Abstract.** We analysed the biogenic volatile organic compound (BVOC) emissions from rapeseed leaf litter and their potential to create secondary organic aerosols (SOA) under three different conditions, i.e., (i) in presence of UV light irradiation, (ii) in presence of ozone, and (iii) with both ozone and UV light. These experiments were performed in a controlled atmospheric simulation chamber containing leaf litter samples, where BVOC and aerosol number concentrations were measured for six days. Our results show that BVOC emission profiles were affected by UV light irradiation which increased the summed BVOC emissions compared to the experiment with solely $O_3$. Furthermore, the diversity of emitted VOCs from the rapeseed litter also increased in presence of UV light irradiation. SOA formation was observed when leaf litter was exposed to both UV light and $O_3$, indicating a potential contribution to particle formation or growth at local scales. To our knowledge, this study investigates, for the first time, the effect of UV irradiation and $O_3$ exposure on both VOC emissions and SOA formation for leaf litter samples. A detailed discussion about the processes behind the biological production of the most important VOC is proposed.

## 1 Introduction

Nowadays, the crucial role played by Volatile Organic Compounds (VOCs) as precursors of ozone and particles within the troposphere has been established (Hatfield and Huff Hartz, 2011). Sources of VOCs are either anthropogenic, related to human activities, or biogenic. Biogenic volatile organic compounds (BVOCs) are released from living and senescent vegetation, soils and microorganisms, or oceans (Kesselmeier and Staudt, 1999; Murphy et al., 2010). Such biogenic VOCs (BVOCs) have been estimated to contribute up to 90% of the total VOC emissions (Guenther, 1995). Furthermore, the currently most accredited emission model for BVOC (MEGAN v2.1) estimates that 760 Tg C yr$^{-1}$ are emitted into the troposphere (Sindelarova et al., 2014). Modelling studies have highlighted the impact of BVOCs on carbon monoxide (CO), hydroxyl radical (OH), and low-level ozone and thus the oxidative capacity of the troposphere (Granier et al., 2000; Pfister et al., 2008; Poisson et al., 2000). It was found that products resulting from the BVOC oxidation are significant precursors of Secondary Organic Aerosols (SOA) that affect the earth's radiative balance (Ziemann and Atkinson, 2012) and thus the climate and human health (De Gouw and Jimenez, 2009). In addition, between 11% and 70% of emitted BVOCs are converted into SOA, leading to a yearly production of 140-190 Tg C yr$^{-1}$ of particles (Hallquist et al., 2009).

Due to the growing awareness about climate change and atmospheric pollution, the number of studies focusing on BVOCs has grown exponentially over the past 20 years, with a strong focus on forests and plants since they are the most important sources of BVOC. However, little attention has been drawn to leaf litter and its contribution to SOA formation in the global BVOC emissions model, even though several studies reported a significant contribution to BVOC emissions, describing BVOCs emitted from leaf litter as potential contributors to SOA formation (Bigg, 2004; Faiola et al., 2014; Isidorov and Jdanova, 2002; Viros et al., 2020). The annual global leaf litter production has been estimated to be between 75 and 135 Pg dry matter (DM) yr-1, contributing to 10% of the global annual emissions of acetone and methanol (Matthews, 1997; Warneke et al., 1999) . It was found that the leaf litter's contribution to acetone and methanol emissions is due to the degradation processes

driven by microorganisms or abiotic factors (i.e., temperature, radiation), processes known to release partially oxidized VOC such as acetone and methanol (Warneke et al., 1999).

Rapeseed (Brassica napus) was chosen as model plant species in this study due to its wide geographic distribution and its importance as a crop. Rapeseed is grown to produce animal feed, edible vegetable oils, and biodiesel. Rapeseed was the third-leading source of vegetable oil in the world in 2000, after soybean and palm oil. It is the world's second-leading source of protein meal after soybean. France is the fifth largest producer worldwide of this specific crop (Fischer et al., 2014).

The development cycle of rapeseed is divided into 3 phases: 1) vegetative, 2) reproduction, and 3) maturation. For the
vegetative phase, rapeseed is sown in August. This phase starts with an epigeous germination during the month of September. From September to December, the rapeseed stem will grow from 10 to 20 cm and produce about 20 leaves forming a rosette. The reproduction phase starts after the winter, i.e., between February and March. It is during this time that the rape goes up. Then we observe the beginning of the elongation. Flowering lasts between 4 and 6 weeks, and the maturation phase occurs when the siliques are formed (in June). In July, they are ready for harvest. It is in this period that we collected the rapeseed
litter.

Rapeseed residues are often left on the field. The incorporation of crop residues into agricultural soils improves soil structure, reduces bulk density, reduces evaporation, and decreases erosion. Rapeseed in this rotation contributes to improving the organic matter content of the soil. Organic matter, which is essential to fertility, contributes to the supply of nitrogen, to the improvement of structural stability (less sensitivity to soil compaction and erosion), and to an increase in the storage capacity
of water and mineral elements (i.e., improvement of the cation exchange capacity) (Tiefenbacher et al., 2021). Therefore, the litter associated to rapeseed is an important aspect of that process.

The volume of straw produced varies between 0.6 and 2.4 tons of dry matter per hectare. This estimate takes into account the important losses of material that occur during mowing operations, and it corresponds to the volume of harvestable straw per hectare. Only half of the total volume produced is harvested, the rest is left in the field to return to the soil (FranceAgriMer,
2016).

The composition and amount of BVOCs emitted from leaf litter, alongside their associated reactivity, strongly depend on plant species, decomposition state, and environmental conditions such as temperature, ultraviolet (UV) light irradiation, and ozone concentration. Nevertheless, ozone concentration in rural areas has been estimated to be around 60 ppb with peaks reaching 80 ppb during the summer (Monks et al., 2015). This affects leaf litter directly through chlorosis and cellular damage (Diaz-
de-Quijano et al., 2016). Also, ozone indirectly impacts biological and chemical processes such as photosynthesis, respiration, stomatal functioning (Yendrek et al., 2017), and the emissions of BVOCs (Yuan et al., 2016, 2017a, b). Another important factor affecting the degradation of leaf litter is UV light (Derendorp et al., 2011) which is responsible for increased emissions of short length VOCs (i.e., C2-C5) especially in the presence of humid air (Derendorp et al., 2011).

This study aims to investigate the individual and combined effects of ozone and UV light irradiation on BVOCs emission and
the subsequent SOA formation from rapeseed litter, *Brassica Napus sp*. Rapeseed litter was used because it is the third most cultivated species in France after wheat and maize (French National Statistics, 2019). We investigated the VOC emission profiles of the senescent rapeseed leaves for 6 days after they were collected. The experiments were carried out in a multiphase simulation chamber where leaf litter was exposed to (i) UV light (UV), (ii) ozone ($O_3$), and (iii) a combination of both (UV_$O_3$).

## 2 Materials and Methods

**2.1 Samples collection**

The leaves of rapeseed (sp. *Brassica napus*) used during the experiments were collected on June 3rd, 2019 in the AgroParisTech field, Thiverval-Grignon (48°85′N, 1°95′E). The Thiverval-Grignon site is located about 30 km west of Paris, in the North of France. The soil of this site is classified as Luvisol. It consists of 25% clay, 70% silt, and 5% sand. The site is

15 ha and the rapeseed leaves were collected using the random sampling method. To avoid inhomogeneous samples in terms of the decomposition stage, all of the leaves were cut directly from the stems but making sure that they were falling or about to fall. Overall, 3 kg of leaves were collected from different plants in the field (field area around 1 km2). The rapeseed litter used for the measurements was made of leaves at the beginning of senescence. The leaf samples were stored for at -20 °C. The sampled litter was reused for all the measurements, throughout the experimentation, defrosting just the fraction of sample needed for the experiment. At the beginning of each experiment, the leaves had visually the same aspect and identical mass to volume ratio (as an indirect metric of their decomposition). In addition, the VOCs were monitored during the stabilisation of the experimental conditions and showed identical patterns.

## 2.2 Samples preparation

The rapeseed leaves were acclimatized for about 2 h at 20 °C before being inserted into the multiphase simulation chamber. In this way, leaves reached room temperature (20 °C), which corresponds to the average temperature in the North of France during summertime. This was necessary for reproduction of the real-time conditions under which the rapeseed leaves start their decomposition. Once acclimatized, leaves were weighted and spread out to cover the whole surface of a FEP (fluorinated ethylene propylene) film (with a surface of 0.64 $m^2$) (**Figure 1a**). After 6 days of measurement, the surface covered by the rapeseed litter was estimated to be 0.45 $m^2$ (**Figure 1b**) using Adobe Photoshop software (V 21.1.1). Photoshop allowed the manual selection of the pixels containing the litter, the pixels were converted into surface area ($m^2$) using the following formula:

$$A_{litter} = \frac{Px_{litter}}{Px \times m^2} \qquad \text{(eq. 1)}$$

where $A_{litter}$ is the area covered by the rapeseed litter 6 days after the beginning of the experiment, $Px_{litter}$ is the number of pixels in the litter area, and $Px \times m^2$ is the number of pixels per $m^2$. The initial weight of rapeseed in the chamber ranged from 75 to 80 g. After 6 days of measurement, the weight decreased by 29-32 %. After being spread on the FEP, the samples were introduced into the multiphase simulation chamber.

## 2.3 Multiphase simulation chamber

The multiphase atmospheric simulation chamber is schematized in **Figure 2**. The atmospheric chamber has a rectangular shape with 1m length $\times$ 1m width $\times$ 2m height (total volume 2 $m^3$). The chamber is made of FEP film. The chamber was continuously filled with 6 L $min^{-1}$ of purified air, where 2 L $min^{-1}$ of this total flow was directed inside a glass bubbler to maintain a constant relative humidity inside the chamber (RH = 50±5 %) (**Figure 2**). The overall air renewal time in the chamber was around 5h 30, which allows for chemical reactions to occur. The chamber was equipped with 12 UV lamps (OSRAM lamps, Eversun L80W/79- R), 6 on the left wall and 6 on the right wall of the chamber. The absolute irradiance within the chamber has been already reported by (Alpert et al., 2017). Light produced from the UV fluorescent tubes had wavelengths between 300 to 400 nm. Alpert et al., (2017) also reported that measurements for $\lambda < 300$ nm yielded detection limit values on the order of $10^{-3}$ W $m^{-2}$ $nm^{-1}$, and thus total light output below 300 nm is negligible. The full spectrum is shown in **Fig. S1** for completeness. In comparison, the solar spectrum at the earth's surface is shown. It was derived using the online Quick Tropospheric Ultraviolet and Visible (TUV) calculator for a solar zenith angle of 0° (available at http://cprm.acom.ucar.edu/Models/ TUV/Interactive_TUV/).

Temperature, relative humidity, and differential pressure (to ensure a slight overpressure in the chamber compared to laboratory air) were monitored using a combined sensor for temperature and relative humidity (Vaisala HUMICAP humidity, and Temperature Probe HMP110; Vaisala Differential Pressure Transmitter PDT101). Data about the monitored temperature are reported in Fig. S3. VOCs and particle formation were monitored using a high-resolution proton transfer reaction mass

spectrometer (PTR-TOF-MS 8000, Ionicon Analytik) and a scanning mobility particle sizer spectrometer (SMPS - model 3080, TSI), respectively.

## 2.4 Experimental set-up

The rapeseed litter was studied within a multiphase simulation chamber. The rapeseed litter was tested under three different conditions to distinguish the potential factors influencing the VOC emissions and the particle formation. The chosen conditions were under (i) UV light irradiation, (ii) ozone, and (iii) ozone and UV light irradiation at the same time. The UV light irradiation was turned off and on following the night/day cycle; the UV light was turned on for a total of seven hours per day. The ozone was injected into the chamber once a day at the same time that the UV light was turned on. The initial concentration of 80 ppb was progressively consumed during the day. Every sample was analysed during 6 days for each of the previously mentioned conditions. **Tab. 1** summarizes the different experimental runs performed in this study. For each of the selected conditions, blank experiments were made for 3 days under the same conditions and subtracted from the following experiments.

## 2.5 Particle measurements

Particles were detected by means of an SMPS consisting of a differential mobility analyser (DMA, model 3085, TSI) and an ultrafine condensation particle counter (UCPC model 3776 high flow, TSI, $d_{50} > 2.5$ nm). During the experiments, the scanning particle size ranged from 2.5 to 79.1 nm, and both the sheath and sample flow rates were settled at 3 and 0.3 L min$^{-1}$, respectively. The SMPS inlet was positioned at 180 cm above the rapeseed surface to observe the particle formation and growth. The density of the measured particles was assumed to be 1 g cm$^{-3}$. The particle loss due to the impact of the chamber walls was calculated based on data from previous experiments performed on the same multiphase simulation chamber (Alpert et al., 2017; Bernard et al., 2016). The estimation of the particle loss used for the correction of the SMPS data are shown in Appendix A, Fig. S2.

## 2.6 VOCs measurement

VOCs were detected using the PTR-TOF-MS technique, which has been already described in detail by Müller et al., (2014). Ionization of the VOCs was carried out using the $H_3O^+$ mode. The pressure and voltage of the drift tube were respectively set to 2.2 mbar and 500 V with a temperature of 80 °C. Consequently, the E/N ratio was about 123 Td (1 Td=$10^{-17}$ V cm$^2$). These parameters were maintained constant during the whole experiment to avoid different ionization conditions of the VOCs within the drift tube. The sample inlet of the PTR-TOF-MS was constantly heated at 60 °C to avoid product loss by absorption in the inlet tube. The instrument sampled every 30 seconds with a flow rate of 100 mL/min, and the raw data was recorded using TofDaq software (Tofwerk AG, Switzerland). The PTR-TOF-MS has a mass resolution of 4500 m/Δm. The PTR-TOF-MS has a mass resolution of 4500 m/Δm. A calibration gas standard (TO-14A Aromatic Mix, Restek Corporation, Bellefonte, USA) containing 14 VOCs at a concentration of 100 ± 10 ppb in nitrogen was used to calibrate and regularly assess the instrument performance, including mass resolution, mass accuracy, sensitivity, and relative mass-dependent transmission efficiency. The sensitivity of these compounds ranged between 15 and 70 cps/ppb, depending on the actual mass. However, since it was not possible to calculate the exact sensitivity for all the detected compounds, we assumed that the proton reaction constant was always equal to $2 \times 10^{-9}$ cm$^3$ s$^{-1}$ (Cappellin et al., 2011; Kalalian et al., 2020), and thus the average sensitivity of 30 cps/ppb was applied for all the compounds. Moreover, the calibration of the spectra was performed via both an oxygen isotope of the ion source $H_3^{18}O^+$ (21.022 m/z) and an ionized acetone molecule $C_3H_7O^+$, (59.0449 m/z) as described by Cappellin et al. (Cappellin et al., 2011). Those compounds were chosen for the calibration because their identification was straightforward for all the kinds of samples used in this study.

After calibrating the spectra, a peak table was created including the largest number of detected compounds. The threshold for the automatic research feature of the peak was set at 0.1 counts per second. Even if the peaks were automatically identified, a manual readjustment of every peak was performed to reduce the bias of the automatic peak research. The range of the detected masses was between 31 m/z and 164 m/z. Masses deriving from the water cluster, such as 37.03 m/z, 38.03 m/z, 39.03 m/z, and 55.03 m/z, were not taken into account during the analysis of the dataset.

Furthermore, the mixing ratio (ppb) was calculated using the PTR-viewer software (V3.2.8, Ionicon, Analytik GmbH) which used the equation described in Cappellin et al., (2011), and the VOC emissions fluxes ($E_{VOC}$) in µg m$^{-2}$ h$^{-1}$ were calculated as follows:

$$E_{VOC} = \frac{F_{air} \times ([VOC]_{litter} - [mVOC]_{blank}) \times M_{VOC}}{V_{mol}^{air} \times ((S_{litter-S} + S_{litter-E})/2) \times 1000 \ (ng/\mu g)} \qquad \text{(eq. 2)}$$

where $F_{air}$ is the net airflow ($F_{air}$ = 240 L h$^{-1}$), $[VOC]_{litter}$ is the concentration (ppb) of the VOC emitted in the chamber with the samples, and $[VOC]_{blank}$ is the concentration (ppb) of the VOC measured in the empty chamber. $M_{VOC}$ is the molecular mass of the corresponding VOC (g mol$^{-1}$), $V_{mol}^{air}$ is the air molar volume at standard temperature and pressure (24.79 L mol$^{-1}$ at 25°C and 1 atm), and $S_{litter-S}$ is the exposed surface of litter to light when the experiment started and $S_{litter-E}$ is the exposed surface of litter to light when the experiment ended.

### 2.6.1 Peaks identification method

The spectra were analysed using the Spectra Analyser tool of the PTR viewer software (Version 3.2.8, Ionicon). This tool allowed identification of the compounds corresponding to each peak in the spectra by searching for the possible combinations of elements leading to the closest molecular weight. This identification of the VOC was also double-checked with literature reviews. Even if this method accounted for the most precise identification of the VOCs, it does not provide a certain identification of the compounds since (1) it is not possible to distinguish between two ion masses that are closer than the PTR-TOF-MS mass resolution, and (2) the PTR-TOF-MS does not distinguish between isomers (VOCs having the same molecular mass).

### 2.6.2 Data analysis

The database was filled with 217 variables corresponding to the number of detected masses for the three different conditions UV, $O_3$, and $UV\_O_3$ (as mentioned previously). The statistical analysis of the entire dataset was performed using the R software (Version 1.2.5019– ©2009–2019 RStudio). At first, we selected the variables that were normally distributed by the Shapiro-Wilk test (W>0.9). Secondly, we tested the homogeneity of the variance by the Levene-test to perform the Analysis of Variance (ANOVA) test followed by the Tukey post-hoc test. Furthermore, we tested the differences between the conditions using the Principal Component Analysis (PCA – package *FactormineR*). The PCA allowed a graphical representation of the whole dataset differentiating the VOCs emission profiles for the different tested conditions without bias. A table with the 30 most emitted compounds and their relative abundance at the three different conditions is presented in **Table 1**. Finally, the calculation of the Shannon index was performed. The Shannon index is a quantitative measure reflecting how many different VOCs were emitted from each sample. It was calculated with the diversity function of the vegan package (version 2.4-3) in the R software (version 3.2.3). The diversity index was calculated as $H = \sum_{VOC} E_{VOC} \log(E_{VOC})$, where the sum is over all VOCs recorded in the mass table.

## 3 Results

### 3.1 Ozone and UV light irradiation effect on the average VOC concentrations

VOC emissions were measured for 6 days for each condition. The summed VOC emissions from the different conditions showed a statistical difference for every day of measurement. Under UV irradiation (the first condition), the summed VOC emissions kept increasing until the 5th day of measurement, while during the last day it statistically decreased (**Figure 3a**). Whereas, for the conditions $O_3$ and UV_$O_3$ (second and third conditions), the summed VOC emissions increased the 2nd day and then slowly decreased from the third to the 6th day of measurement (**Figure 3b** and **3c**).

Furthermore, the ANOVA test confirmed a difference between the averages of the summed VOC emissions per day. These results highlight a statistical increase of the summed VOC emissions under UV irradiation (first condition) and a statistical decrease of the summed VOC emissions over time for the $O_3$ and the UV_$O_3$ condition. The summed VOC emissions were higher for the UV condition than for the UV_$O_3$ condition. The condition with the lowest VOC emission rate was with ozone.

The VOC emission profiles of the different conditions are compared in Figure 4. The PCA shows that the VOC profiles emitted during the UV condition were separated from the VOC profiles emitted from the UV_O3 and O3 conditions. Meanwhile the UV_$O_3$ and the $O_3$ conditions had very similar profiles since their ellipses are superposed (Figure 4). The major differences in the emission profiles were led by the different concentrations of 10 compounds at the following m/z: 45.03, 45.99, 46.03, 47.02, 49.99, 59.049, 60.05, 73.03, 108.95, and 125.95. Those compounds are also among the 30 most emitted compounds through all three conditions (Table 1). The identification of the 30 most emitted compounds for the three different conditions is listed in Appendix A, Table S1.

The 30 most emitted compounds represented 90% of the summed VOC emissions for each condition. The list of the most emitted compounds between the $O_3$ condition and UV_$O_3$ conditions was similar, especially in terms of the types of emitted compounds. The three most emitted compounds for these two conditions were methanol ($CH_3OH^+$, 33.03 m/z), acetaldehyde ($C_2H_4OH^+$, 45.03 m/z), and butyric acid ($C_4H_8O_2H^+$, 89.05 m/z). While for the UV condition, the three most emitted VOC were acetic acid (61.03 m/z), acetone (59.049 m/z), and methanol (33.03 m/z). The average contribution of the VOCs over the 6 measurement days showed a large difference for each condition. For instance, methanol contributed to 9%, 32 %, and 50%, for UV, $O_3$, and UV_$O_3$, respectively.

### 3.2 Evolution and diversity of the VOC emissions per day

For the UV light experiments, small changes were observed. For example, the average contribution of acetic acid (m/z 61.03) increased between 10-15 % during days 3 to 5 compared to days 1, 2, and 6, while that of methanol (33.03 m/z) increased by 5% during days 2 and 3. However, the other most emitted VOCs contribution was constant during that time (**Figure 5a**). For the $O_3$ condition, the most important change in the average contribution is represented by the mass 33.03 m/z, which increased by 7% between the 4th and the 6th day (**Figure 5b**). It is also worth mentioning that masses 42.03 m/z and 49.99 m/z contributed to less than 0.01% of the total VOC emissions during the 1st and 2nd day of measurements, while after 3 days their contribution increased by an 80 and 200-fold change, reaching 0.8% and 2% of the VOC relative abundance, respectively. The average contribution of the mass 89.06 m/z decreased over time from 5% during the first day to 0.5% during the 6th day of measurements. Similar behaviour was reported for the mass 73.06 m/z, where its average contribution increased to 8% during the 2nd and 3rd measurement day and then decreased to 0.5% of the average contribution during the 6th day. Analysing the UV_$O_3$ condition, we noticed that the variation in the VOC contribution per day is higher than for the other conditions. In addition, the results in **Figure 5c** reported that the mass 71.05 m/z was strongly emitted during the 6th day of measurement (30%), whereas this VOC emission did not reach 0.01 % of the contribution in the previous days. The average contribution of the mass 33.03 m/z decreased over time, passing from 70% on the 1st and 2nd day to 30% on the 6th day of measurements.

Moreover, the Shannon Index, representing the diversity of emitted VOC, was calculated for each day of measurements to
highlight an increase or a decrease of the VOC diversity with time. The VOC Shannon index showed that there were no
statistical differences in terms of VOC diversity that were observed for the UV light condition (S.I. 3.05 – 3.28) and $O_3$
condition (p value >0.05). Concurrently, the UV_$O_3$ condition results showed a statistically significant increase of the VOC
diversity with time (from 1.54 to 2.4). The Shannon index of the VOC also showed a significantly larger Shanonn index for
the UV condition compared to the UV_$O_3$ condition (3.15 compared to 2). An intermediate value of 2.35 was obtained for the
$O_3$ condition.

**3.3 Ozone and UV light irradiation effect on particle formation**

Concurrently with the detection of VOC emissions, we also investigated particle formation for the three different conditions.
Under UV irradiation, nucleation started 1 hour after switching on the UV light (**Figure 6a**). The initial nucleation produced
a dense number of particles between $5x10^4$ and $8x10^4$ particles $cm^{-3}$. Then, the number of particles decreased, while their
diameter increased from 2 nm to 40 nm. Likewise, under the ozone condition (**Figure 6b**), a nucleation event also started 1 h
after the injection of 80 ppb of ozone. However, compared to the UV light irradiation experiment, the ozone injection led to a
lower number of particles formed ($2.5x10^4$ particles $cm^{-3}$) with a smaller diameter (<17 nm). Nevertheless, when the UV light
irradiation was combined with ozone injection (third condition), the nucleation was stronger than the first two cases, reaching
a maximum of $3.5x10^5$ particles $cm^{-3}$ for particle diameters between 2 and 12 nm (**Figure 6c**). Ozone depletion was also faster
than in the case where only $O_3$ was used (i.e., condition two).

Furthermore, **Figure 7** regroup the 8 VOC was positively correlated with temperature for the UV_$O_3$ condition. The positive
correlation means that VOC emissions increased with the temperature. Those compounds have a Spearman coefficient higher
than -0.80. For the other VOC not displayed in **Figure 7,** correlations lower than 0.8 were found.

**4 Discussion**

**4.1 UV light and Ozone affect the diversity of the VOC emission profiles.**

For the $O_3$ and UV_$O_3$ experiments, the VOC diversity decreased while the methanol contribution increased. Potard et al.,
(2017) observed similar behaviours in their experiment, which consisted of measuring VOC emissions from soils receiving
different types of amendment: the highest methanol average contribution corresponded to the lowest VOC diversity. Moreover,
differentiated VOC profiles have been highlighted in the PCA (**Figure 4**) between the UV light experiment and the $O_3$ and
UV_$O_3$ experiments. Several mechanisms are regulating the VOC emissions and thus affecting the VOC diversity. These
mechanisms are discussed in detail in the following paragraphs.

**4.2 Effect of ozone and UV light irradiation on the most emitted compounds**

**Acetic acid ($C_2H_4O_2$, 61.03 m/z) and formic acid ($CH_2O_2$ 47.02 m/z).** Organic acids such as acetic and formic acid are
mostly emitted from living plants (Kesselmeier and Staudt, 1999) and from the foliage of trees and crops with a flux of 35 µg
$m^{-2}$ $h^{-1}$ (Paulot et al., 2011). Viros et al., (2021) also detected acetic and formic acid from senescent litter with a flux of 0.05
and 0.98 µg $m^{-2}$ $hr^{-1}$, respectively. In this study, the emission rates of the two organic acids ranged from 0.76 to 64.28 µg $m^{-2}$
$h^{-1}$ for acetic acid, and from 0.23 to 9.12 µg $m^{-2}$ $h^{-1}$ for formic acid. Mozaffar et al., (2018) described that the acetic acid
emissions were affected by temperature, as they recorded lower emissions in the early morning than during the late afternoon.
This could explain the higher emissions of acetic acid observed in our study where the temperature reached 30 °C, which is
higher than the conditions encountered by Viros et al., (2021) i.e., 22 °C. Mozaffar et al., (2018), while analysing BVOCs from
senescent maize leaf litter, reported an acetic acid contribution to the total BVOC emission of up to 26 %. Similar results were

obtained in our study where the contribution of the acetic acid reached 20% of the total VOC emissions under UV light conditions.

**Methanol** ($CH_4OH^+$, 33.03 m/z). Methanol was the most emitted compound in $O_3$ and $UV\_O_3$ conditions. Methanol emission from plants is ubiquitous (Bracho-Nunez et al., 2011; Gonzaga Gomez et al., 2019; Harley et al., 2007; Wiß et al., 2017). Moreover, methanol is the most emitted VOC from crops and other plants such as *Cistus albidus, Coronilla valentina, and Prunus persica* (Harley et al., 2007), and it often contributes to more than half of the overall VOCs emissions. Hence, in our study, the average methanol contribution to the total VOCs emission is between 8.9 % (under UV) and 50 % (under UV and $O_3$). Gonzaga Gomez et al., (2019) measured VOC emissions from rapeseed using dynamic chambers and reported that methanol contributed from 56 to 77% of the summed VOC emissions. These values are higher than in the current study. The reason behind this difference could be that Gonzaga Gomez et al., (2019) measurements were performed over the whole growing plant, while in our experiment we only analysed the emission from the leaf litter. Furthermore, the emissions of methanol from leaves depend on the phenological stage of the plant (Wiß et al., 2017), which could be another factor differentiating this study from that of Gonzaga Gomez et al., (2019). In fact, in this study, we measured mature leaves in the last phenological state, while Gonzaga Gomez et al., (2019) analysed leaving plants in the flowering and grain filling stages. Mature leaves are known to emit less methanol than young ones (Harley et al., 2007). Methanol is produced via the demethylation of the pectin by Pectin Methyl Esterase (PME) activity. This process occurs during cell wall growth which is an intense process happening during the early stage of leaf expansion (Fall and Benson, 1996). Comparing the results obtained for the $UV\_O_3$ condition with that of Harley et al., (2007), where their experimental conditions were the closest to those used here, we found that the methanol emissions were in the same range. The emission flux of methanol under the $UV\_O_3$ condition in the current study is $0.22 \pm 0.03$ µg $g^{-1}$ $h^{-1}$, while Harley et al., (2007) reported fluxes ranging from 0.2 to 2.7 µg $g^{-1}$ $h^{-1}$ for mature leaves. Moreover, under the UV condition, our results show a higher emission rate of methanol, which is in line with previous studies that demonstrated how the UV light increased the methanol emissions from leaves (Derendorp et al., 2011; Harley et al., 2007). Greenberg et al., (2012) detected a methanol flux of 1.3 µg $m^{-2}$ $h^{-1}$ from litter corresponding to 0.4 % of the total emission above the canopy, estimated to be 300 µg $m^{-2}$ $h^{-1}$. In this study, the methanol flux from leaf litter ranged from 4.6 to 28.4 µg $m^{-2}$ $h^{-1}$ depending on the experimental conditions. Hence, our results suggest that the contribution to the total above canopy methanol emissions of the rapeseed litter could range from 2 to 10%.

**Acetaldehyde** ($C_2H_4OH^+$, 45.03 m/z). Acetaldehyde was the second most emitted compound for the $O_3$ condition and the 3rd and the 4th most emitted for the $UV\_O_3$ and UV experiments, respectively. In general, the mechanisms leading to acetaldehyde emissions are still uncertain. The most accredited hypothesis is that these emissions are correlated with different types of stress such as ozone exposure and leaf damage (chlorosis) caused by the sunlight (Seco et al., 2007). In this study, leaves were under high ozone concentration (60-80 ppb) and intense UV irradiation, which could have accelerated the senescence period of the rapeseed leaves inside the chamber. As a result of these stressing conditions, we obtained larger acetaldehyde emissions than in previous studies. For instance, Greenberg et al., (2012) reported a VOC flux for leaf litter under the canopy of 0.3 µg $m^{-2}$ $h^{-1}$, corresponding to 0.2% of the total above canopy acetaldehyde emissions, while in this study the emission flux ranged from $1.97 \pm 0.01$ µg $m^{-2}$ $h^{-1}$ for the $UV\_O_3$ condition to $26.7 \pm 0.2$ µg $m^{-2}$ $h^{-1}$ for the UV condition. The total above canopy acetaldehyde emissions reported by Greenberg et al., (2012) were 200 µg $m^{-2}$ $h^{-1}$. As for methanol, our study suggests a higher contribution (ranging from 2 to 13%) to the total above canopy acetaldehyde emissions from leaf litter.

However, Hörtnagl et al., (2014) reported a burst of 1900 µg $m^{-2}$ $h^{-1}$ after a meadow cutting. Nonetheless, another pathway for the production of acetaldehyde is ethanol oxidation at the leaf level, forming acetaldehyde (Niinemets et al., 2014; Seco et al., 2007). This process only occurs in anaerobic conditions since it is the consequence of the ethanolic fermentation pathway. Hence, acetaldehyde can be formed in leaf tissues, but this pathway cannot be the main reason for the acetaldehyde emissions detected in this study because the leaf litter was not in an anoxic environment. The magnitude of the acetaldehyde emission rate detected is similar to the one detected by Bachy et al., (2016) from soil hosting C4 crops ($7 \pm 9$ µg $m^{-2}$ $_{soil}$ $h^{-1}$). Therefore,

we underline the possibility that rapeseed leaf litter might contribute to tropospheric acetaldehyde emissions at the same level as soil and plants under environmental stress conditions.

**Acetoin** ($C_4H_8O_2H^+$, 89.06 m/z). Acetoin was the second and the third most emitted compound for the conditions UV_$O_3$ and $O_3$ respectively with an average contribution to the summed VOC emissions between 9 and 11%. This compound has already been reported as one of the most emitted compounds from bacteria dwelling in rapeseed samples (Wagner et al., 2018). These bacteria have been identified as *Enterobacter*, *Klebsiella*, *Serratia*, *Staphylococcus*, and *Streptomyces* (Schulz and Dickschat, 2007). The pyruvate metabolic pathway of the microorganisms just listed allows the production of the acetoin molecule by the decarboxylation of acetolactate (Schulz and Dickschat, 2007). The large production of this compound can be attributed to the presence of bacteria colonizing the leaves' surfaces and also to the favourable conditions for bacteria growth, such as the optimal temperature (T= 25 °C) (Membre et al., 2005) and humid atmosphere (RH= 50%) (Mceldowney and Fletcher, 2008) in our experiments.

**Acetone ($C_3H_6OH^+$, 59.049 m/z).** This compound was largely emitted from litter under UV irradiation. The average contribution of acetone was 13% under UV light, 1.64 % when influenced by both UV and ozone, and 2 % when the litter was exposed to ozone only. Acetone has been reported to be one of the most emitted compounds by plants and litter (Gonzaga Gomez et al., 2019; Greenberg et al., 2012). Greenberg et al., (2012) reported an average flux of 0.3 µg m$^{-2}$ h$^{-1}$ between 11:00 and 17:00. In this study, the emissions of acetone were 10 times higher under UV irradiation. Based on (Greenberg et al., 2012), the current estimates of litter contribution to the above canopy acetone emissions is 0.1 %. However, the flux reported in Table 2 suggest that the litter contribution to acetone emission, in the absence of ozone, could be as large as 6 %.

In the study of Gonzaga Gomez et al.,(2019), where the VOC detection has been performed at a different phenological stage of the rapeseed plant, acetone was detected among the most emitted VOCs from leaves and was correlated with sunlight because the highest emission peak of acetone occurred at midday. These findings are in line with the higher emissions of acetone in the UV light experiment but not with the UV_$O_3$ experiment. Cojocariu et al., (2005) found that under stress conditions such as high $O_3$ concentration, acetone concentration increased in *Fagus selvatica*. This is in contrast with the results of this study where the $O_3$ concentration seems to reduce the acetone emissions. The biogenic nature of the source of acetone cannot be confirmed since, as reported by Das et al., (2003), acetone emissions could be the result of photochemical reactions of other VOCs. Decaying and senescing plants may be another direct source of acetone (Warneke et al., 1999; Jacob et al., 2002; Karl et al., 2003).

### 4.2.1 Other emitted compound

**Isoprene (C5H8H+, 69.07 m/z).** In this study, isoprene was the 30[th] most emitted compound only in the experiment without $O_3$. Its average contribution in the UV light experiment was 1% with a flux rate of 3.00±0.03 µg m$^{-2}$ h$^{-1}$ or 0.02 µg g$^{-1}$ h$^{-1}$ which is almost 20 times lower than the emissions reported by Morrison et al., (2016), where the maximum detected flux of isoprene from rapeseed was 0.35 µg g$^{-1}$ h$^{-1}$. This difference is probably due to the different samples. Indeed, Morrison et al., (2016) investigated branches, while here only the emissions from senescent leaves were considered. However, the flux rate of isoprene reported by this study is in line with those reported by Gonzaga Gomez et al., (2019), i.e., 0.035 µg g$^{-1}$ h$^{-1}$. Isoprene can also be emitted from microorganisms such as bacteria and fungi. Isoprene is an intermediate product of the mevalonate pathway which leads to the production of essential organic compounds within the microorganism's cells (Hess et al., 2013). Isoprene is therefore a metabolite directly related to the presence of microorganisms in soil and plants (Hess et al., 2013)

### 4.3 Temperature effect on the BVOC emissions

Higher temperatures increase chemical reaction rates, cellular diffusion rates, and the vapor pressure of the VOCs. As a consequence, BVOC emission rates are dependent on temperature. In this study, we identified 8 VOCs emitted from rapeseed

litter which were highly correlated with temperature. Among the most correlated ones, we identified methanol and MEK, in agreement with previous reports investigating such temperature dependence from rapeseed plants (Gonzaga Gomez et al., 2019). Harley et al., (2007) detected methanol emissions from 6 different plant species. Their results reported a correlation between its emission and the temperature of the leaves and stomatal conductance. The mechanisms behind this behaviour have been explained by Niinemets and Reichstein, (2003). Methanol is produced within the cell walls, and it diffuses in the liquid phase following the diffusion gradient until it reaches the surface of the cell walls. Then, methanol diffuses in the gas phase into the substomatal cavity and is released as VOC in the ambient air through the stomata. In our study, stomata lock-open as a consequence of cellular death (Prats et al., 2006), and the increased temperature accelerated the diffusion process releasing methanol as the most emitted compound from rapeseed leaf litter.

## 4.4 SOA formation from leaf litter BVOC emissions

To our knowledge, an investigation of SOA formation from leaf litter samples has only been reported by the study of Faiola et al., (2014). Faiola et al., (2014) reported the maximum peak volume of SOA particles obtained through the oxidation of the emitted VOCs by the injection of 130 ppb of $O_3$ under controlled atmospheric conditions. The experiment was similar to the one performed here where only 80 ppbs of ozone were injected ($O_3$ condition). Comparing the $O_3$ experiment in this study with the experiment under the dry conditions of Faiola et al., (2014) (**Table 2**), the maximum volume of SOA particles in our study has the same order of magnitude as the volume reported by Faiola et al., (2014). The most important difference between this study and the previous one is the concentration of the monoterpenes detected. In Faiola et al., (2014) monoterpenes contributed to 80% of the total VOC emissions. Monoterpenes, together with isoprene and sesquiterpenes, are considered to be the three primary classes of VOCs forming SOA (Sakulyanontvittaya et al., 2008). Isoprene is the most emitted compound from vegetation (Sindelarova et al., 2014) with a relatively small aerosol yield (Henze and Seinfeld, 2006). On the other hand, monoterpenes have been known to widely contribute to SOA formation (Griffin et al., 1999). In this study, monoterpenes were found to be lower than our PTR-TOF-MS detection limit, and isoprene was only the 30[th] most emitted compound under UV light irradiation. For instance, furfural has been reported as a precursor of SOA formation, with an aerosol yield ranging from 0.3 to 3% depending on the ozone concentration (Colmenar et al., 2020). Acetaldehyde and acetone have been reported to be taken up into the aerosol phase and to participate in the aerosol-phase reactions (Barsanti and Pankow, 2004). Those reactions generate products with a relatively low vapor pressure, which leads to an additional partitioning from the gas phase, increasing the organic particulate mass (Limbeck et al., 2003; Tong et al., 2006). In this study, acetaldehyde and acetone were found to be correlated with SOA formation from rapeseed leaf litter and to be largely emitted, from 60 to 40 and from 17 to 12 ppb respectively, in the UV_$O_3$ condition,. The observed particles formation highlighted the high oxidation potential of the UV light irradiation with a volume of particle production per day higher than the one found for the $O_3$ experiment (**Table 2**). Moreover, the combination of ozone and UV light produced a larger maximum aerosol volume peak than the one reported in Faiola et al., (2014) for both dry and wet conditions and the largest aerosol volume per day compared to the $O_3$ and UV light experiments (**Table 2**).

Furthermore, we observed particles in the range from 2.5 to 79.1 nm, while Faiola et al., (2014) detected them between 20 and 730 nm. In this study, for the $O_3$ experiment, the percentage of particles under 20 nm contributed to 38% of the total aerosol volume (**Table 2**). Therefore, aerosol formation from leaf litter was certainly underestimated in this previous study due to the importance of particles below 20 nm.

## 4.5 Conclusion

This study highlighted the possibility that VOC emissions from rapeseed leaf litter, one of the three most cultivated crops in France and worldwide, could have been underestimated. We reported substantial SOA formation for the different studied conditions. In the experiment with UV and $O_3$, the aerosol volume measured in the chamber was 790 $\mu m^3\ cm^{-3}$. It is important

to stress that these results may correspond to lower limits for SOA production since (i) the UV lamps had about seven times lower light intensity at 365 nm than actual solar radiation, and (ii) the detection of the particles was performed up to 79.1 nm, consequently, the formation of particles having greater diameters was not detected. We, therefore, suggest that SOA formation from leaf litter may have an atmospheric impact. This study also highlights the need for further studies to quantify the possible impact of the SOA formation from leaf litter at a larger scale.

In this work, we detected the VOC from rapeseed litter samples for 6 days under three different conditions: UV light irradiation, ozone injection, and UV light combined with ozone injection. The experiments were performed under controlled conditions within an atmospheric simulation chamber. The results showed that BVOC emissions from senescent rapeseed litter impact SOA formation and that the combination of UV light irradiation and ozone injection increased the BVOC emission profile's diversity. UV light irradiation was found to affect the production of SOA more than the $O_3$ injection. In the presence of both UV light and $O_3$, SOA formation was 9 and 52 times higher than from solely UV light or ozone, respectively.

Low emissions of isoprene were detected, even though the production of SOA was not negligible. The densest portion of particles produced by litter samples had a diameter lower than 20 nm, which might have caused an underestimation of the SOA formation from litter in other studies that detected a range of particles with a diameter higher than 20 nm.

**Acknowledgments**

This work received funding from the Agence de la transition écologique (ADEME) under the Cortea program (RAVISA grant agreement No. 1762C0006) and the European Commission (EC) under the European Union's Horizon 2020 research and innovation program (Eurochamp 2020 grant agreement No. 730997).

**Author contribution**

**Abis** L., conceptualization, data curation, investigation, formal analysis, methodology, visualization, writing – original draft preparation. **Kalalian** C. methodology, investigation, writing – review & editing. **Lunardelli** B., methodology, investigation. **Wang** T., **Zhang** L., and **Chen** J. data curation, investigation. **Perrier** S., investigation, methodology, resources. **Loubet** B., writing – review & editing, investigation. **Ciuraru** R., conceptualization, data curation, methodology, writing – review & editing, funding acquisition, project administration, supervision, validation. **George** C., methodology, writing – review & editing, funding acquisition, project administration, supervision, validation.

**Competing interests**

The authors declare that they have no conflict of interest.

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

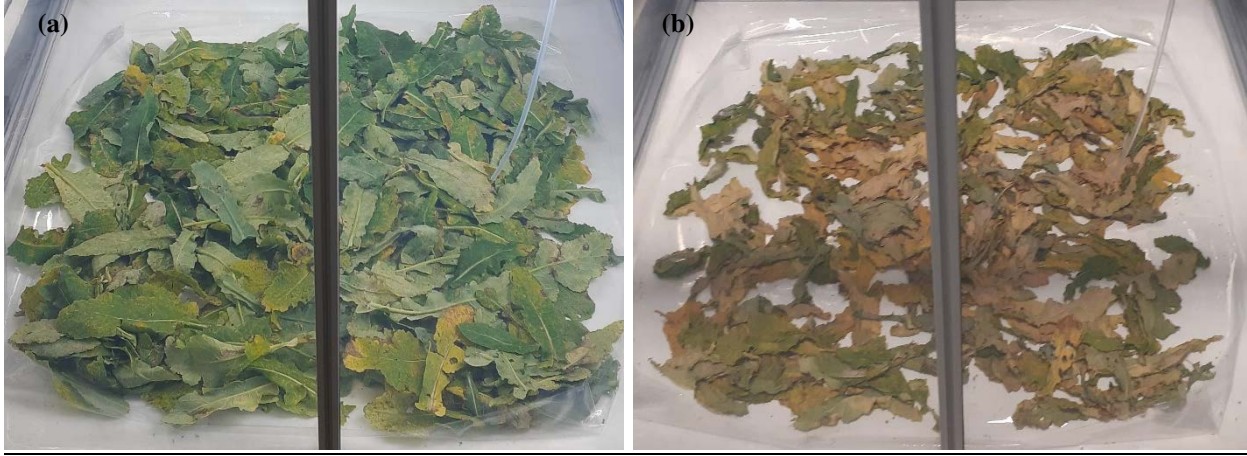

**Figure 1: Example of the Rapeseed litter condition a) during the first day of the VOC and particles measurements and b) after 6 days of VOC and particles measurements.**

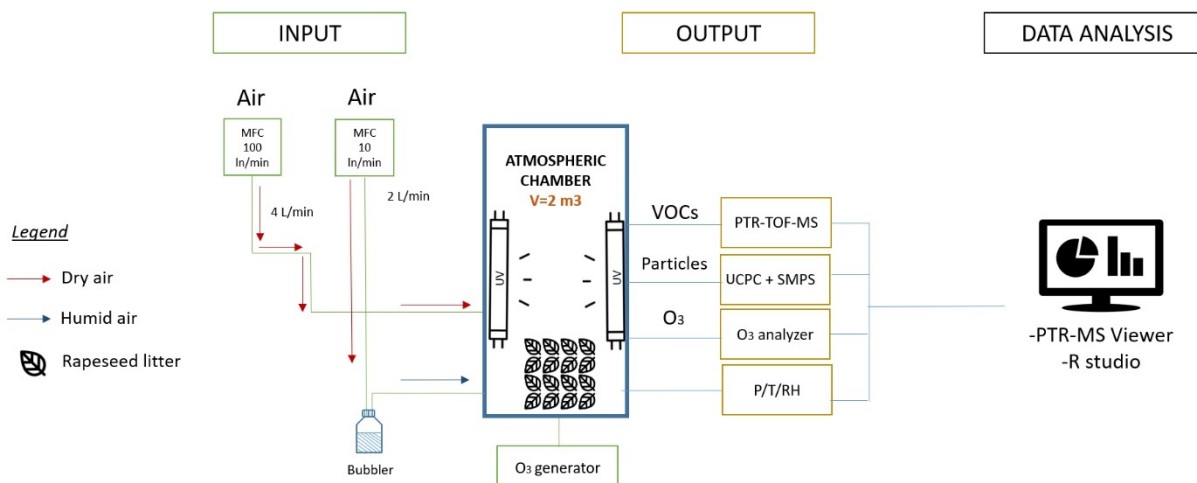

**Figure 2: Scheme of the multiphase reaction chamber used for the study of the photoreactivity of the VOCs emitted from senescence rapeseed. The PTR-TOF-MS have been used for the VOCs detection, the ultrafine condensation particle counter (UCPC) and the SMPS have been used for the detection of the particle formation and measure the size particles, the O₃ analyzer detected the ozone inside the chamber, where P= pressure, T=temperature, and RH=relative humidity has been constantly monitored during the entire experiment.**

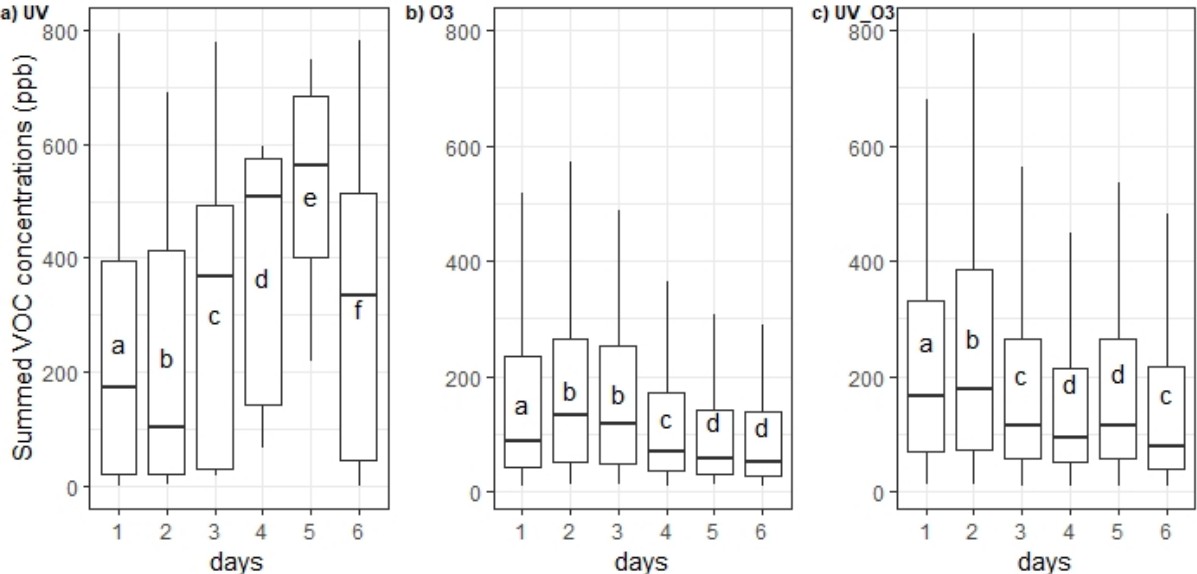

**Figure 3: Summed VOC concentrations for each day (24-h period)incubation condition a) UV, b) O3 and c) UV_O3 Letters indicate the statistical difference obtained by the Tukey test.**

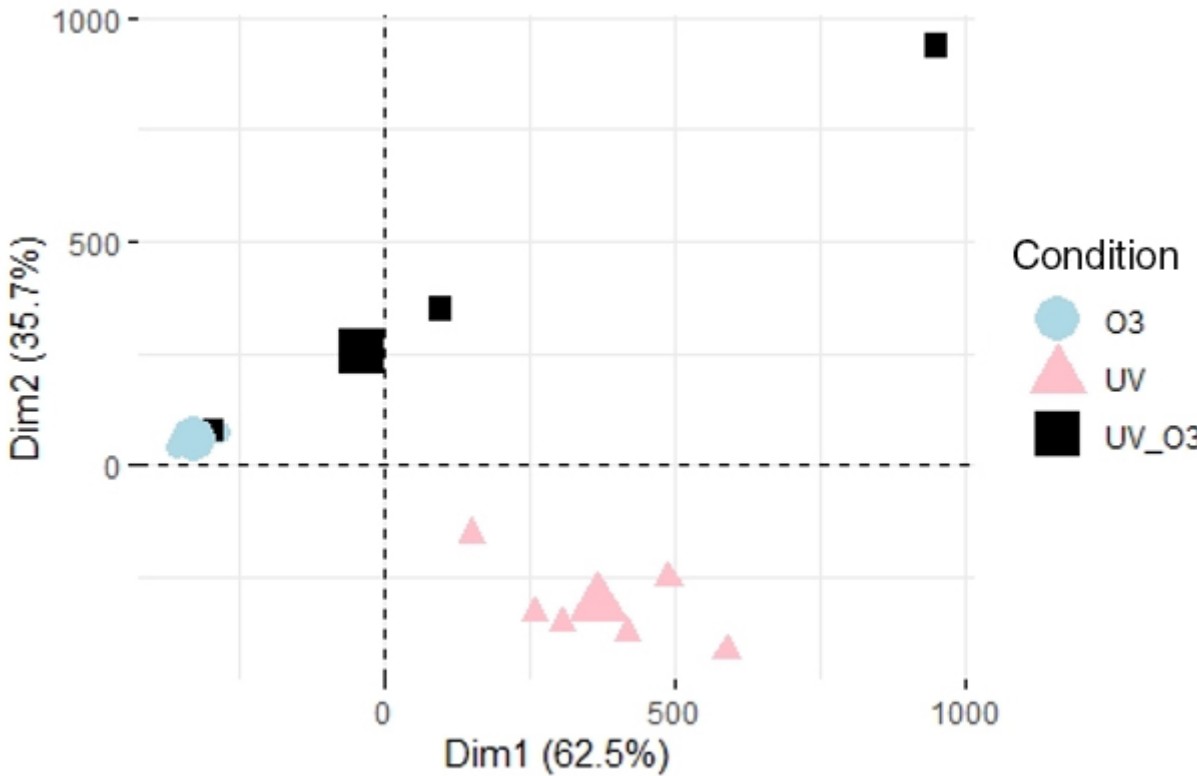

**Figure 4: VOC profiles differences between UV light, UV_O₃, and O₃ conditions, each point represent 1 day measurement. The percentage of the variance explained by the 2 first components is shown on each axis (Dim1 and Dim2).**

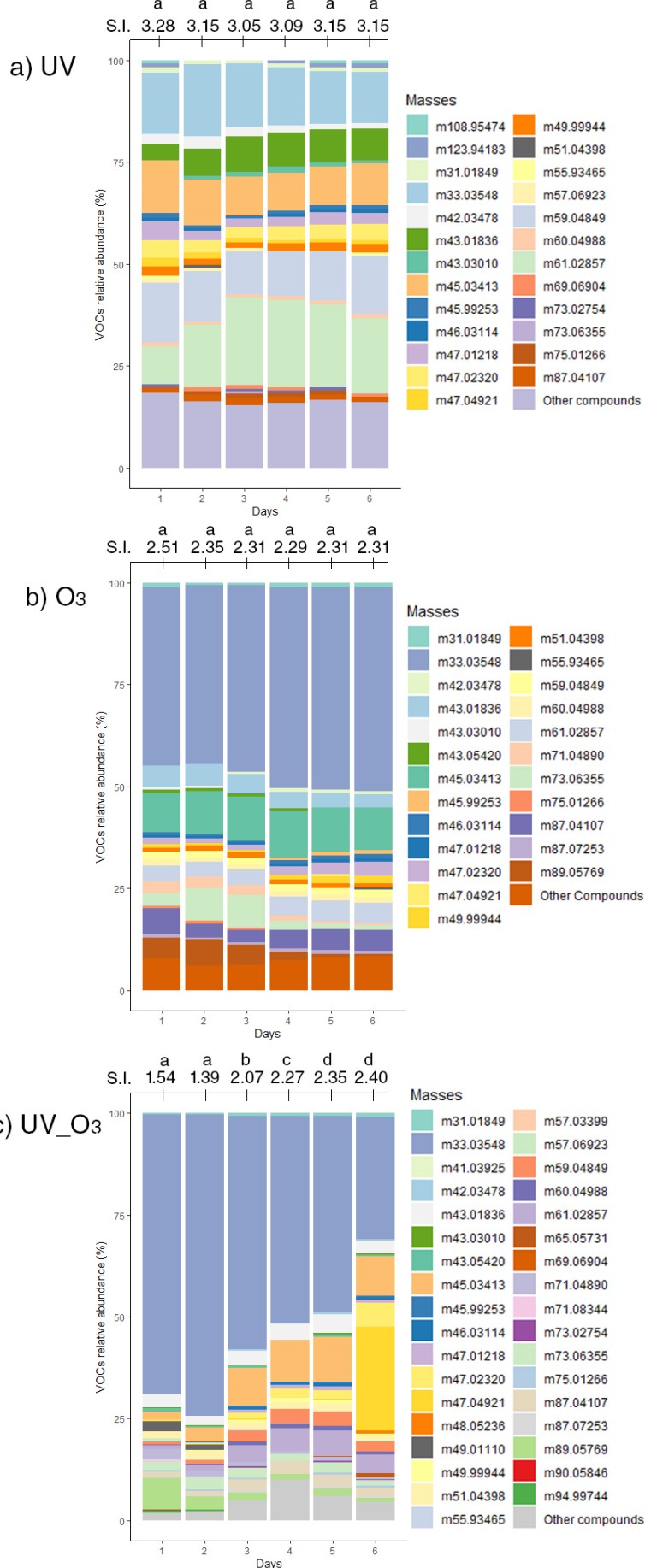

**Figure 5: VOC relative abundance for rapeseed litter samples under a) UV light b) O₃ and c) UV_O₃ conditions. S.I. is the Shannon index representing the diversity of the VOC (for each day). Letters indicate significant differences of the S.I. according to the Tukey test with p.value < 0.05.**

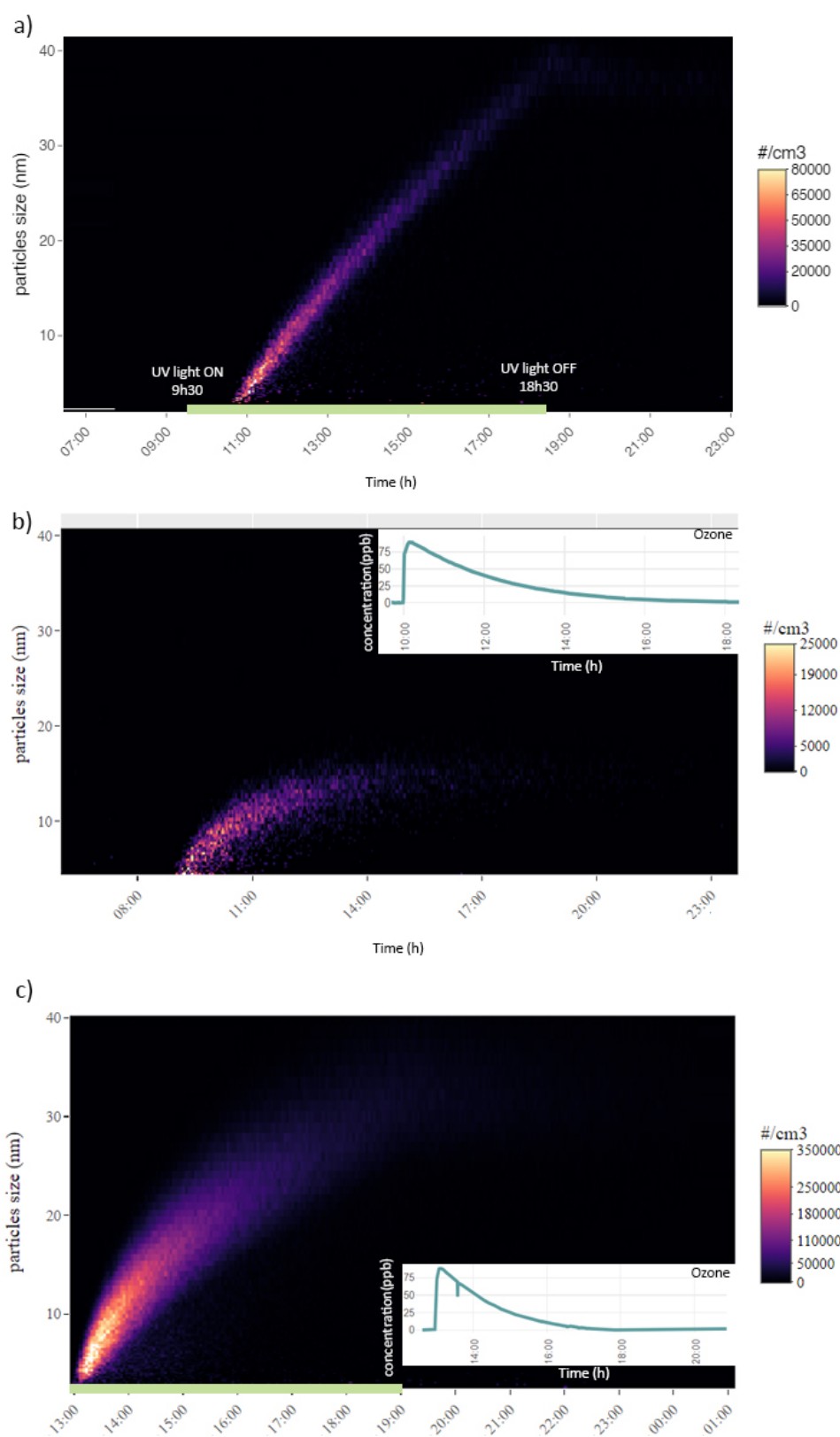

**Figure 6. Temporal evolution of particle number and size distribution, ordinate represents the electrical mobility diameter (nm) and the colour scale the particle number concentration. Particle formation for the first day of measurement under a) UV light irradiation, b) Ozone injection and c) UV light irradiation and ozone injection combined. The green horizontal line represents the timeline where the UV light were switched on, for a) the UV light have been turned on at 9h30 and turned off at 18h30, for c) the UV light have been turned on at 12h30 and turned off at 19h. b) and c) also display the Ozone concentration timeline during the particle formation.**

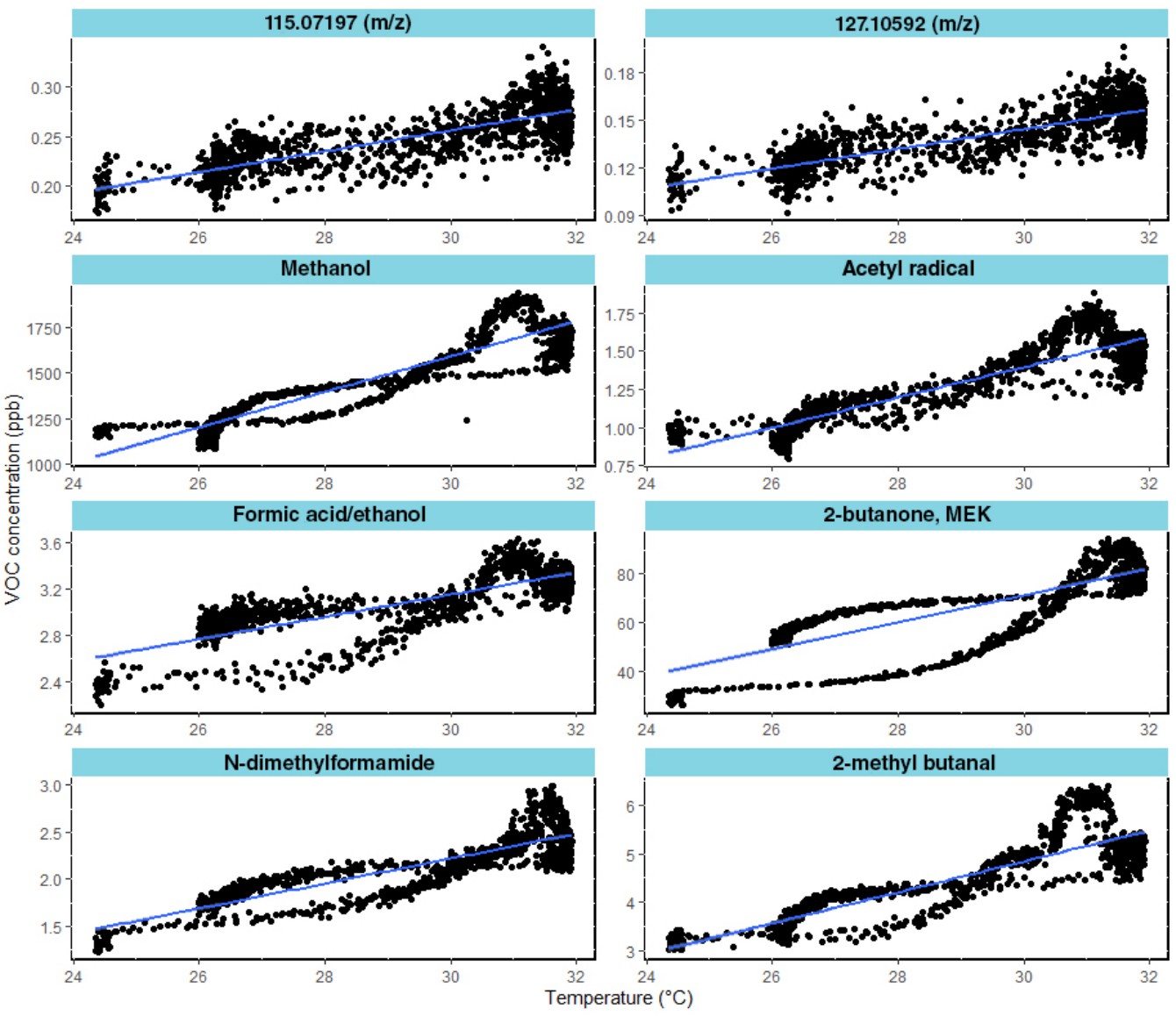

**Figure 7 Correlation between VOC mixing ratios and temperature under the UV_O₃ condition. The 8 most correlated VOC are shown(Pearson correlation coefficients > 0.8). .**

**Table 1. experimental conditions**

|  | Experimental conditions | Colza weight | Surface covered | Days of VOC detection | Days of SOA detection | Blank conditions |
|---|---|---|---|---|---|---|
| UV light | 7h per day of irradiation with UV | Initial weight: 85 g, Weight after 6 days 52 g | Initial surface covered: 0.64 $m^2$ Surface covered after 6 days: 0.45 $m^2$ | 6 | 1 | 3 days averaged with 7h per day of irradiation with UV |
| Ozone | Initial concentration of ozone injected in the chamber: 80 ppbs | Initial weight: 80 g, Weight after 6 days 49 g | Initial surface covered: 0.64 $m^2$ Surface covered after 6 days: 0.45 $m^2$ | 6 | 1 | 3 days averaged with an initial concentration of ozone injected in the chamber of 80 ppbs |
| UV light and ozone | Initial concentration of ozone injected in the chamber: 80 ppbs, 7h per day of irradiation with UV | Initial weight: 80,7 g, Weight after 6 days 47 g | Initial surface covered: 0.64 $m^2$ Surface covered after 6 days: 0.45 $m^2$ | 6 | 1 | 3 days averaged with an initial concentration of ozone injected in the chamber of 80 ppbs and 7h per day of irradiation with UV |

**Table 2. The average of 30 most emitted compound during the 6 days of measurement for the three different conditions: UV light irradiation, Ozone, and UV light irradiation and ozone at the same time. The flux was calculated using the averaged surface of the leaf litter between the initial covered surface (0.64 m²) and the final covered surface (0.45 m²). Within the columns m/z, the compounds highlighted as the most differentiating between the VOC profiles by the PCA are in bold. A tentative identification of the compound here listed is reported in Appendix B, Table B1.**

| MOST CONCENTRATED COMPOUNDS (UV) | | | MOST CONCENTRATED COMPOUNDS (O₃) | | | MOST CONCENTRATED COMPOUNDS (UV_O₃) | | |
|---|---|---|---|---|---|---|---|---|
| m/z | BVOC Flux (µg/m²/h)* ± sd | Average contribution (%) | m/z | BVOC Flux (µg/m²/h)* ± sd | Average contribution (%) | m/z | BVOC Flux (µg/m²/h)* ± sd | Average contribution (%) |
| 61.03 | 64.28±0.60 | 20.16 | 33.03 | 4.59±0.01 | 31.52 | 33.03 | 24.35±0.09 | 50.40 |
| **59.049** | 41.50±0.06 | 13.01 | **45.03** | 1.44±0.01 | 9.89 | 89.06 | 5.24±0.001 | 10.85 |
| 33.03 | 28.38±0.17 | 8.90 | 89.06 | 1.16±0.001 | 7.96 | **45.03** | 1.97±0.01 | 4.09 |
| **45.03** | 26.71±0.20 | 8.38 | 73.06 | 1.15±0.001 | 7.92 | 73.06 | 1.93±0.00 | 3.99 |
| 43.02 | 18.35±1.90 | 5.76 | 87.04 | 1.10±0.08 | 7.58 | 87.04 | 1.53±0.42 | 3.18 |
| **47.02** | 9.12±1.20 | 2.86 | 61.03 | 0.76±0.01 | 5.21 | 71.04 | 1.48±0.76 | 3.07 |
| 87.04 | 7.78±1.17 | 2.44 | 43.02 | 0.61±0.04 | 4.21 | 43.02 | 1.46±0.22 | 3.02 |
| 47.01 | 7.23±1.66 | 2.27 | 71.05 | 0.46±0.03 | 3.16 | 61.03 | 1.26±0.02 | 2.60 |
| 123.94 | 5.41±0.08 | 1.70 | **59.049** | 0.30±0.001 | 2.07 | 51.04 | 1.02±0.001 | 2.12 |
| **49.99** | 5.34±0.02 | 1.68 | **47.02** | 0.23±0.02 | 1.60 | 49.01 | 0.92±0.001 | 1.90 |
| 42.03 | 5.18±0.47 | 1.63 | 60.04 | 0.22±0.01 | 1.50 | **59.049** | 0.79±0.001 | 1.64 |
| **108.95** | 3.75±0.10 | 1.18 | 51.04 | 0.19±0.001 | 1.32 | 47.05 | 0.65±0.06 | 1.34 |
| 75.01 | 3.69±0.05 | 1.16 | 87.07 | 0.18±0.02 | 1.23 | 94.99 | 0.39±0.16 | 0.81 |
| 47.05 | 3.13±0.08 | 0.98 | **49.99** | 0.14±0.001 | 0.95 | **47.02** | 0.35±19.23 | 0.73 |
| 69.07 | 3.00±0.03 | 0.94 | 75.01 | 0.14±0.001 | 0.94 | 57.07 | 0.35±0.001 | 0.73 |
| **60.05** | 2.83±0.30 | 0.89 | **47.01** | 0.09±0.03 | 0.64 | 71.08 | 0.34±0.001 | 0.71 |
| **73.03** | 2.79±1.04 | 0.87 | 43.05 | 0.09±0.01 | 0.61 | 43.05 | 0.33±0.02 | 0.67 |
| 43.03 | 2.59±0.34 | 0.81 | **46.03** | 0.09±0.03 | 0.60 | 87.07 | 0.28±0.07 | 0.58 |
| 101.06 | 2.59±0.41 | 0.81 | 42.03 | 0.07±0.001 | 0.49 | **60.05** | 0.27±0.01 | 0.56 |
| 87.07 | 2.30±0.20 | 0.72 | 31.01 | 0.07±2.22 | 0.47 | 75.012 | 0.26±0.001 | 0.55 |
| 45.99 | 2.30±0.33 | 0.72 | **45.99** | 0.07±0.02 | 0.45 | 90.06 | 0.15±0.001 | 0.32 |
| 73.06 | 2.27±0.04 | 0.71 | 43.03 | 0.06±0.06 | 0.43 | **47.01** | 0.15±0.65 | 0.31 |
| **125.95** | 2.25±0.07 | 0.71 | 123.94 | 0.06±0.001 | 0.42 | 31.02 | 0.14±0.46 | 0.30 |
| 90.95 | 2.16±0.06 | 0.68 | 47.05 | 0.05±0.001 | 0.35 | **49.99** | 0.13±0.001 | 0.27 |
| 57.06 | 2.12±0.08 | 0.67 | 88.04 | 0.05±0.01 | 0.34 | 43.03 | 0.12±0.60 | 0.25 |
| 55.93 | 1.85±0.06 | 0.58 | **73.03** | 0.05±1.26 | 0.34 | 46.03 | 0.11±0.05 | 0.23 |
| **46.03** | 1.84±2.19 | 0.58 | 55.93 | 0.05±0.001 | 0.32 | 123.94 | 0.09±0.001 | 0.18 |
| 57.03 | 1.78±0.74 | 0.56 | 90.06 | 0.05±0.01 | 0.31 | 42.03 | 0.08±0.04 | 0.16 |
| 31.01 | 1.70±0.21 | 0.53 | 74.06 | 0.05±0.11 | 0.31 | 74.06 | 0.07±0.53 | 0.15 |
| 93.95 | 1.64±0.10 | 0.51 | **108.95** | 0.04±0.001 | 0.28 | 96.007 | 0.07±0.001 | 0.14 |

**\*conversion to µg/g$_{DM}$/h can be obtained by substituting the averaged surface with the g of dry matter (45g)**

**Table 3. Comparison of the SOA formation from leaves litter samples reported in this study and the literature.**

| Sample type | Sampling period | Measured particles range (nm) | Type of chamber | Experimental conditions | Maximum peak of aerosol formation ($\mu m^3$ $cm^{-3}$) | Total Aerosol volume concentration ($\mu m^3$ $cm^{-3}$) | Volume Contribution of particles < 20 nm | Ref. |
|---|---|---|---|---|---|---|---|---|
| Mix of: Pinus ponderosa, Pseudotsuga menziesii Pinus monticola, Larix occidentalis litter and soil | May-June 2012 | 20-730 | atmospheric chamber (7.7 m³) | 130 ppb of $O_3$ in dry conditions | 0.97-5.43 | - | - | (Faiola et al., 2014) |
| Mix of: Pinus ponderosa, Pseudotsuga menziesii Pinus monticola, Larix occidentalis litter and soil | May-June 2012 | 20-730 | atmospheric chamber (7.7 m³) | Reproducing raining event 130 ppb of $O_3$ | 0.29-2.55 | - | - | (Faiola et al., 2014) |
| Brassica napus litter | June 2019 | 2.5-79.1 | Multiphase simulation chamber (2m³) | 60-80 ppb of $O_3$ | 0.2 | 15.1 | 38% | This study |
| Brassica napus litter | June 2019 | 2.5-79.1 | Multiphase simulation chamber (2m³) | Only UV light | 0.8 | 85.4 | 24% | This study |
| Brassica napus litter | June 2019 | 2.5-79.1 | Multiphase simulation chamber (2m³) | 60-80 ppb of $O_3$ and UV light | 7.6 | 787.8 | 24% | This study |