# Peer review of "Measurement Report: Biogenic VOC emission profiles of rapeseed leaf litter and its SOA formation potential"

_Atmospheric Chemistry and Physics, 2021_

## Author Response (AR1)

Villeurbanne, le 16 juin 2021

**Jason D. Surratt**
**Atmospheric Chemistry and Physics**
**Editor**
**Special Issue: Simulation chambers as tools in atmospheric research (AMT/ACP/GMD inter-journal SI)**

*Title: Measurement report: Biogenic VOC emissions profiles of Rapeseed leaf litter and their SOA formation potential*
*Author(s): Letizia Abis et al.*
*MS No.: acp-2021-135*
*MS type: Measurement report*

Dear Jason,

We are hereby submitting our revised manuscript entitled "Biogenic VOC emission profiles of rapeseed leaf litter and its SOA formation potential" by Abis et al.

We are most thankful to the reviewers who raised a series of relevant points, helping us improving our manuscript.

As it can be seen in the point-to-point answer below, we took into account most, if not all, points raised by the reviewers in our revised manuscript, which has also been proof read by a professional reader.

I believe that this revised version now meets ACP's criteria as a measurement report, and hope it will be accepted.

I do need however to stress that this submission is belonging to a special issue supported by a European project, namely Eurochamp-2020, which ends in August. If eventually you accept this manuscript for publication, it is then important for us that the page charge are invoiced before the end of the project.

Yours sincerely,

Christian GEORGE

**IRCELYON ● UMR 5256**
**2 avenue Albert Einstein ● F-69626 Villeurbanne Cedex**
**tél. +33 [0] 472 431 4 89 ● fax: +33 [0] 472 448 438**
**Christian.George@ircelyon.univ-lyon1.fr ● http://www.ircelyon.univ-lyon1.fr/**

[Figure]

**Reviewer 1**

This paper present the emission of biogenic volatile organic compounds (BVOC) from rapeseed leaves litter under three different experimental conditions i.e. under UV light irradiation, in presence of ozone, and under simultaneous exposure to ozone and UV light irradiation. The experiments were carried out in a simulation chamber containing leaves litter collected nearby Paris in north of France.

The most emitted compound was methanol followed by acetaldehyde, acetoin and acetone in O3 and UV-O3 conditions. Surprisingly, isoprene was the 30th most emitted compound only in the experiment without presence of O3. The BVOC emission influenced the secondary organic aerosol (SOA) formation process. In the presence of both UV light and O3 the SOA formation was 9 and 52 higher than only UV light or ozone.

To my opinion this manuscript can be of broad interest for the atmospheric chemistry community, and it can be published in ACP. I have few comments that can possibly improve the quality of the manuscript prior to be published.

We would like to thank this reviewer for his/her insightful and helpful comments. They have helped us to improve our manuscript. Hereafter, please find our answers.

**Main comments:**

- In the "Experimental procedure" it is not clear how many experiments were performed (it is ambiguous for the blank experiments and missing for the experiments themselves). The authors should clearly state upon how many replicates are based their conclusions and provide a table for various initial conditions and main results.

Obviously, the rapeseed litter is seasonal, and is limited due to its collection procedure. On the day of the collection, the rapeseed litter used for the measurements was made of leaves at the beginning of the senescence process. Replicate experiments are difficult to be performed in this case since other samples would have a different degree of senescence and therefore difficult to compare with the first set of experiments. The evolution of the litter over time is accompanied by a change in the colour of the leaves from green to yellow to brown. This is due to a degradation of the metabolism leading to the death of the cells and the degradation of the chlorophyll. To repeat the experimentation, we would need to renew the litter samples the following year.

Nevertheless, we had obviously to define an experimental plan to address the scientific questions underlying to this work. Such a procedure increases the reproducibility of the starting material for each runs performed here (in total nine runs). We initially performed a preliminary study (not included in our manuscript) where the BVOC emission and SOA formation from rapeseed litter was investigated in the presence of both UV light and ozone (100 ppb). This testing showed some reproducibility (with some inherent variability when working with biological samples). We then decided to perform further experiments under complementary conditions (i.e., $O_3$, UV light, or both), to see the impact of each parameter on the BVOC emission and SOA formation. For each condition, the experiments were repeated 2 times. Therefore, the BVOC data are the average of these replicas. However, due to a SMPS failure, only one replica by condition was available.

Table 1 below summarizes the different experimental runs performed in this study. For each selected conditions, blank experiments were made for 3 days under the same conditions and subtracted from the following experiments.

*Table 1. List of experiments performed (each repeated twice), according to sample weight and surface covered*

| | Experimental conditions | Colza weight | Surface covered | Days of VOC detection | Days of SOA detection | Blank conditions |
|---|---|---|---|---|---|---|
| UV light | 7h per day of irradiation with UV | Initial weight: 85 g, Weight after 6 days 52 g | Initial surface covered: 0.64 m$^2$ Surface covered after 6 days: 0.45 m$^2$ | 6 | 1 | 3 days averaged with 7h per day of irradiation with UV |
| Ozone | Initial concentration of ozone injected in the chamber: 80 ppbv | Initial weight: 80 g, Weight after 6 days 49 g | Initial surface covered: 0.64 m$^2$ Surface covered after 6 days: 0.45 m$^2$ | 6 | 1 | 3 days averaged with an initial concentration of ozone injected in the chamber of 80 ppbv |
| UV light and ozone | Initial concentration of ozone injected in the chamber: 80 ppbv, 7h per day of irradiation with UV | Initial weight: 80,7 g, Weight after 6 days 47 g | Initial surface covered: 0.64 m$^2$ Surface covered after 6 days: 0.45 m$^2$ | 6 | 1 | 3 days averaged with an initial concentration of ozone injected in the chamber of 80 ppbv and 7h per day of irradiation with UV |

- Sometimes, the analysis are oversimplified. Some key measurements are not given, and the literature survey is not wide enough.

We added new data our measurements such as temperature, relative humidity, and pressure inside the chamber. Also, we expanded the literature survey on Rapeseed and associated VOC emissions, which now reads as:

*Rapeseed (Brassica napus) was chosen in this study as model plant species due to its wide geographic distribution and its importance as a crop. Rapeseed is grown for the production of animal feed, edible vegetable oils, and biodiesel. Rapeseed was the third-leading source of vegetable oil in the world in 2000, after soybean and palm oil. It is the world's second-leading source of protein meal after soybean. France is the fifth producer worldwide of this specific crop (Fischer et al., 2014).*

*The development cycle of rapeseed is divided into 3 phases: 1) the vegetative; 2) the reproduction and 3) the maturation. For the vegetative phase, rapeseed is sown in August. This phase starts with an epigeous germination during the month of September. From September to December, the rapeseed stem will grow from 10 to 20 cm and about*

*20 leaves forming a rosette. The reproduction phase, starts after the winter i.e., between February and March. It is at this time that the rape goes up. We observe then the beginning of the elongation. Flowering lasts between 4 and 6 weeks and the maturation phase is when the siliques are formed (in June). In July, they are ready for the harvest. It is in this period that we collected the rapeseed litter.*

*Rapeseed residues are often left on the field. The incorporation of crop residues into agricultural soils improves soil structure, reduces bulk density, reduces evaporation, and decreases erosion. Rapeseed in this rotation contributes improving the organic matter content of the soil. Organic matter, which is essential to fertility, contributes to the supply of nitrogen, to the improvement of structural stability (less sensitivity to soil compaction and erosion), and to the increase in the storage capacity of water and mineral elements (i.e., improvement of the cation exchange capacity) (Tiefenbacher et al., 2021). Therefore, the litter associated to Rapeseed is an important aspect of that process.*

*The volume of straw produced varies between 0.6 and 2.4 tons of dry matter per hectare. This estimate takes into account the important losses of material that occur during mowing operations and it corresponds to the volume of harvestable straw per hectare. Only half of the total volume produced is harvested, the rest is left in the field to return to the soil (FranceAgriMer, L'Observatoire National des Ressources en Biomasse (ONRB): Evaluation des ressources disponibles en France ; 2016).*

The experimental section was also revised, in order to include more information.

- The authors should have tried to better define the behaviour of the chamber walls toward the air/light system. This is a valuable exercise which is required for most of the chamber application. There is no information about the estimated water quantity adsorbed on the Teflon wall or about the VOCs adsorbed on the wall.

The dynamic chamber used in this study was made fluorinated ethylene propylene film, FEP. This material is inert and is widely used in the literature in the analysis of BVOC from vegetation (Peron et al., 2021; Timkovsky et al., 2014). This measurement report did not aim at providing a full characterization of the chamber, but rather to provide new insights on the emissions from Rapeseed litter. This chamber have been already documented in the literature (Alpert et al., 2017; Bernard et al., 2016). But obviously, this reviewer is correct, wall effects are always important, and previous research has shown that time scales for organic monoacids to equilibrate with Teflon chamber walls occurs on the order of minutes ( Krechmer, et al., 2016). Therefore, experimental time scales here allowed for equilibrium adsorption-desorption conditions. However, the chamber was run in a dynamic (i.e., flow) mode where the chamber was continuously flushed with an airflow to compensate the air withdrawn by the various analysers connected to it. Such conditions limited, while not removing them, the impact of wall losses and lead to fast response times of the chamber due to external stimuli (light, ozone, etc.). In addition, at the end of each experiment, the chamber was scrubbed using ethanol, then rinsed with water and dried thoroughly before each experiment. Under such conditions and by monitoring the blank levels of VOCs, we can state that the wall conditions were constant and maintained at a low level.

- Is the temperature constant during the chamber experiments?

The temperature raised from 26 to 31 C during the experiment with UV lights switched on. The temperature was however constant for the dark experiments with $O_3$ only. We now added figures showing the temperature variations for all three experimental conditions in Appendix - A.

**Minor comments**

- As the wall material seems to have a significant importance, please provide the precise reference of the material: producer, ref number, and product name.

The chamber is made of fluorinated ethylene propylene film, FEP, obtained from Katco UK (Dupont FEP, 100μm thick).

- As the Teflon foil (FEP) is new and used just before the preliminary experiments how the blank experiments were distributed during the campaign? If, they were evenly distributed among experiments, did you notice any evolution of the wall chemical behaviour?

The blank was recorded during three days before each experiment. At the end of each experiment, the chamber, has been cleaned up with ethanol and flushed with an air flow of 60L/min to avoid any background effect from the previous experiment. We can say, looking at the evolution of the VOC concentration of the blank, that if any evolution of the chemical behaviour happened was negligible. The chamber is made of fluorinated ethylene propylene film, FEP, obtained from Katco UK (DuPont FEP, 100μm thick).

- Adsorbed organics on the chamber wall can also come from the foil production process.

As stated above, the background level of VOC was monitored through the various blank measurements and the chamber regularly cleaned (see above). Under such conditions, the impact of walls was maintained at a low level.

The section "Atmospheric Implications" and "Conclusion" can be combined as they are both very short or strengthen the "Atmospheric Implications" with some examples.

We agree with this reviewer, we merged both sections.

**References**

Alpert, P.A., Ciuraru, R., Rossignol, S., Passananti, M., Tinel, L., Perrier, S., Dupart, Y., Steimer, S.S., Ammann, M., Donaldson, D.J., George, C., 2017. Fatty Acid Surfactant Photochemistry Results in New Particle Formation. Sci Rep 7, 12693. https://doi.org/10.1038/s41598-017-12601-2

Bernard, F., Ciuraru, R., Boréave, A., George, C., 2016. Photosensitized Formation of Secondary Organic Aerosols above the Air/Water Interface. Environ. Sci. Technol. 50, 8678–8686. https://doi.org/10.1021/acs.est.6b03520

Fischer, R.A., Byerlee, D., Edmeades, G., 2014. Crop yields and global food security: will yield increase continue to feed the world? ACIAR Monograph No. 158. Australian Centre for International Agricultural Research. Canberra.

Je, K., D, P., Pj, Z., Jl, J., 2016. Quantification of Gas-Wall Partitioning in Teflon Environmental Chambers Using Rapid Bursts of Low-Volatility Oxidized Species Generated in Situ. Environ Sci Technol 50, 5757–5765. https://doi.org/10.1021/acs.est.6b00606

Peron, A., Kaser, L., Fitzky, A.C., Graus, M., Halbwirth, H., Greiner, J., Wohlfahrt, G., Rewald, B., Sandén, H., Karl, T., 2021. Combined effects of ozone and drought

stress on the emission of biogenic volatile organic compounds from *Quercus robur* L. Biogeosciences 18, 535–556. https://doi.org/10.5194/bg-18-535-2021

Tiefenbacher, A., Sandén, T., Haslmayr, H.-P., Miloczki, J., Wenzel, W., Spiegel, H., 2021. Optimizing Carbon Sequestration in Croplands: A Synthesis. Agronomy 11, 882. https://doi.org/10.3390/agronomy11050882

Timkovsky, J., Gankema, P., Pierik, R., Holzinger, R., 2014. A plant chamber system with downstream reaction chamber to study the effects of pollution on biogenic emissions. Environ Sci Process Impacts 16, 2301–2312. https://doi.org/10.1039/c4em00214h

**Reviewer 2**

**Review Summary**

Abis et al. present measurements of biogenic volatile organic compound (BVOC) emissions from rapeseed leaf litter. Leaves were collected in the field, transported to the lab, and placed in an FEP chamber where they were exposed to one of three conditions: 1) UV irradiance, 2) 80 ppb once-daily ozone injection, or 3) UV irradiance + 80 ppb once-daily ozone injection. UV lights were turned on and off to represent a 7 hour daytime light schedule. BVOC emissions were measured continuously with a PTR-MS for 6 days. In addition, an SMPS was used to monitor particle formation from oxidation of the BVOC emissions in the chamber. The paper highlights this as a potential significant source of secondary organic aerosol (SOA). The topic is interesting and worthy of investigation. However, the limited number of replicates of each condition preclude any ability to make meaningful comparisons. Furthermore, the analysis is described at a superficial level that reads as an early draft, but still requires additional data synthesis and interpretation before publication. I recommend rejection at this stage, but encourage the authors to increase their replicates (or at least better discuss the implications of their results within the context of their limited replicates) and to synthesize the data more thoroughly to complete the project. I provide some ideas for how to proceed with data analysis below.

**General Comments**

In general, there were a lot of grammatical errors that made the manuscript very difficult to read. I recommend sending it to an editing service. Some examples include: referring to "leaves litter" instead of the correct, "leaf litter" throughout the text; writing "biogenic volatile organic compounds emissions" instead of the correct, "biogenic volatile organic compound emissions"; capitalizing terms that are not proper nouns, such as "Volatile Organic Compounds"; L. 28 "Furthermore, the currently most accredited emission model for BVOC (MEGAN v2.1), estimates that 760 Tg C yr-1 are emitted into troposphere"; L. 49 "This affect leaves litter"; "Samples collection" and "Samples preparation" instead of the correct, "Sample collection" and "Sample preparation"; "leaves have been weighted" instead of the correct, "leaves have been weighed". These are just some examples. Not an exhaustive list. I also recommend using "were" or "was" instead of "have been" or "has been" throughout the methods section. It would make it much easier to read.

We would like to thank this reviewer for his/her insightful and helpful comments. They helped us to significantly improve our manuscript. We also do apologies for the poor English in the initial submission. The revised version will be thoroughly edited for English grammar. Hereafter, please find our point-to-point answer to the comments raised.

The number of replicates for each condition were not stated anywhere in the methods. Based on what is written (and what is missing), I assume there was only a single 6-day experiment conducted with leaf litter under each condition (UV, O3, UV_O3). This makes it impossible to compare between the different conditions because we have no information about the natural variability between different leaf litter samples under the same laboratory conditions. I highly recommend conducting more replicates to explore natural variability between samples under the same experimental conditions. If that is not possible, the authors could present this instead as a survey of the change over time in emissions and SOA formation from each condition, separately, BUT it is not appropriate to make comparisons between the conditions when N=1.

In this study, we investigated the influence of the several factor (such as light or ozone) on the emissions of BVOCs from the considered litter. We are presenting a set of chamber runs under those changing experimental conditions, which is quite standard for this kind of experiments (as it can easily be seen in this special issue). A number of preliminary runs have been performed, using small quantities of samples, to set the best conditions for our investigation. We agree with this reviewer that multiplying the experimental runs under varying conditions is required to improve the accuracy of the data obtained. While our strategy is quite common for chamber runs, we faced here another difficulty, which is related the samples investigated.

Obviously, the rapeseed litter is seasonal, and is limited due to its collection procedure. On the day of the collection, the rapeseed litter used for the measurements was made of leaves at the beginning of the senescence process. The same litter is reused for all the measurements throughout the experimentation to study the behaviour of the VOCs emitted over time. Replicate experiments are difficult to be performed in this case since other samples will have a different degree of senescence and difficult to compare with the first set of experiments. The evolution of the litter over time is accompanied by a change in the colour of the leaves from green to yellow to brown. This is due to a degradation of the metabolism leading to the death of the cells and the degradation of the chlorophyll. To repeat the experimentation, we need to renew the litter samples the following year.

Nevertheless, we had obviously to define an experimental plan to address the scientific questions underlying to this work. Such a procedure increases the reproducibility of the starting material for each runs performed here (in total 9 runs). We initially performed a preliminary study (not included in our manuscript) where the BVOC emission and SOA formation from rapeseed litter was investigated in the presence of both UV light and ozone (100 ppb). This testing showed some reproducibility (with some inherent variability when working with biological samples). We then decided to perform further experiments under complementary conditions (i.e., $O_3$, UV light, or both), to see the impact of each parameter on the BVOC emission and SOA formation. For each condition, the experiments were repeated 2 times. So, the BVOC data are the average of these replicas. However, due to a SMPS failure, only one replica by condition was available.

The analysis presented was preliminary. I highly recommend adding some additional simple box modeling to better interpret the chemistry occurring in the chamber. Models such as GECKO could provide a place to start. Furthermore, to make any statements about the potential regional impact of these results on SOA formation, the authors should provide more detailed estimates of how much SOA the leaf litter BVOCs could contribute and how this compares to typical ambient measurements. At the moment, the authors have not made a compelling case that this could actually be a significant source of SOA.

Well, we do need to disagree somehow with this reviewer here, and turn this comment into a request of providing more insights into the gas phase chemistry of emitted BVOCs. We however fully realize that our study does not cover all aspects the chemical transformations occurring in or on the litter, in the gaseous and particulate phases. It rather aims at uncovering the above-mentioned influences of light or ozone on the emissions pattern, with a focus on SOA. We do need to stress that models, such as a Gecko, would not be able to address the complex heterogeneous (on the litter), multiphase (in the particles) and gaseous homogeneous chemistries. Going into more details on all those aspects certainly goes far beyond one single study. This work provides more qualitative insights into a subfield of atmospheric chemistry where data are sparse. The format and content of this submission therefore match nicely the selected manuscript type i.e., a measurement report. Such type concerns new results from measurements of atmospheric properties and processes from field and laboratory experiments, with conclusions of more limited scope than in research articles.

The authors do not provide proper context for using rapeseed leaf litter as an important system for studying this topic. Even if it is the third most commonly cultivated species in France, don't agricultural crops contribute to a minor fraction of total leaf litter in France? And how would agricultural land management practices influence the leaf litter? Do rapeseed leaves senesce every year? What time of year? If so, what do the farmers usually do with that litter? Do they just leave it on the ground for natural decomposition or do they manage it? For example, do they remove the litter once the leaves senesce from the branches? What implications does this have for regional impacts? This does not provide a compelling rationale to study rapeseed litter for this project and there is some missing information that would help us understand the broader context of these results.

Rapeseed (Brassica napus) was chosen in this study as model plant species due to its wide geographic distribution and its importance as a crop. Rapeseed is grown for the production of animal feed, edible vegetable oils, and biodiesel. Rapeseed was the third-leading source of vegetable oil in the world in 2000, after soybean and palm oil. It is the world's second-leading source of protein meal after soybean. France is ranked at the fifth producer worldwide for this specific crop (Fischer et al., 2014).

The development cycle of rapeseed is divided into 3 phases: 1) vegetative; 2) reproductive and 3) maturation. For the vegetative phase, rapeseed is sown in August. This phase starts with an epigeous germination during the month of September. From September to December the rapeseed stem will grow from 10 to 20 cm and about 20 leaves forming a rosette. The reproduction phase, starts after the winter i.e., between February and March. It is at this time that the rape goes up. We observe then the beginning of the elongation. Flowering lasts between 4 and 6 weeks and the maturation phase is when the siliques are formed (in June). In July, they are ready for the harvest. It is in this period that we collected the rapeseed litter.

Rapeseed residues are often left on the field. The incorporation of crop residues into agricultural soils improves soil structure, reduces bulk density, reduces evaporation, and decreases erosion. Rapeseed in this rotation contributes improving the organic matter content of the soil. Organic matter, which is essential to fertility, contributes to the supply of nitrogen, to the improvement of structural stability (less sensitivity to soil compaction and erosion), and to the increase in the storage capacity of water and mineral elements (i.e., improvement of the cation exchange capacity) (Tiefenbacher et al., 2021). Therefore, the litter associated to Rapeseed is an important aspect of that process.

The volume of straw produced varies between 0.6 and 2.4 tons of dry matter per hectare. This estimate takes into account the important losses of material that occur during mowing operations and it corresponds to the volume of harvestable straw per hectare. Only half of the total volume produced is harvested, the rest is left in the field to return to the soil (FranceAgriMer, L'Observatoire National des Ressources en Biomasse (ONRB) :Evaluation des ressources disponibles en France ; 2016).

As consequence, bearing in mind the general importance of Rapeseed as a major crop, and the associated important litter, the new knowledge gained by our study comes with some regional importance that can indeed only be fully addressed by some mesoscale modelling. However, such modelling is currently quite difficult due to our limited knowledge. Therefore, we judged our investigations worthy to be published as measurement report.

**Specific Comments**

L. 48: 60 ppb rural background ozone seems REALLY high. Perhaps, double-check this number and better clarify what this means. Is this the annual average? A daytime average?

A particular rural area that is affected by a nearby city? This is much higher than a typical background mixing ratio of tropospheric ozone.

We checked these data. They do correspond to a rural location where the sampled were collected.

Data from the Airparif network (© 2021 Airparif, https://www.airparif.asso.fr/) reported an average concentration of ozone in this rural area, North-West to Paris, area of 60 µg/m$^3$, corresponding to 30 ppb, for the year 2020. The maximum ozone concentration, in the same area, reached 216 µg/m$^3$, corresponding to 108 ppb of ozone. Furthermore, 27 days with a concentration higher than 60 ppb were observed in 2020, and the peak ozone concentration was reached in June. All these data are reported in the "2020 annual report" of the Airparif network as a free data source.

L. 70: authors state "leaves reached room temperature, which corresponds to the average temperature in the north of France during summertime". Which is what temperature, approximately? The actual temperature itself should be stated here.

We thank the reviewer for the comment. We revised this sentence; it now reads as:

In this way, leaves reached room temperature (20 °C), which corresponds to the average temperature in the north of France during summertime.

L. 81: authors state that the weight of the leaves decreased by 29-32% after the 6-day experiment. How much of this loss is just water? This should be mentioned. Otherwise, the implication here seems to be that this much mass of VOCs was released, which I suspect was actually a minor component of the loss of mass.

The weight loss was dominated by water. This will be stressed in the revision version.

L. 90: it is fine to only show the detailed spectrum of the lamps in the SI, but some general information about the lamps should still be included in the main text. For example, what range of wavelengths does it emit? How does this compare with UV exposure in an ambient environment?

We thank the reviewer for the comment, we added this information in the revised experimental section, which now reads as:

The absolute irradiance within the chamber has been already reported by (Alpert et al., 2017). Light produced from the UV fluorescent tubes had wavelengths between 300 to 400 nm. Alpert et al., (2017) also reported that measurements for λ < 300 nm yielded detection limit values on the order of $10^{-3}$ W m$^{-2}$ nm$^{-1}$, and thus total light output below 300 nm is negligible. The full spectrum is shown in **Fig. A1** for completeness. In comparison, the solar spectrum at the Earth's surface is shown derived using the online Quick Tropospheric Ultraviolet and Visible (TUV) calculator for a solar zenith angle of 0° (available at http://cprm.acom.ucar.edu/Models/ TUV/Interactive_TUV/).

L. 97: authors state the multiphase simulation chamber "allowed the closest representation of the atmospheric conditions." This statement needs a lot more context. What does this mean, "closest representation to atmospheric conditions"? By what metric? By temperature, light, humidity? Are the UV lamps actually similar to the UV the leaves would experience in the field? Were the experiments seeded with polydisperse seed aerosol? If not, the surface area to volume ratio of this chamber could certainly lead to substantial wall loss of oxidized VOC vapors. This is also different from "atmospheric conditions". It is fine to be different from atmospheric conditions, but this statement should be qualified with the ways in which the

chamber represents the natural environment well AND the ways in which the chamber likely does NOT represent the natural environment very well. This helps provide necessary context for interpreting the results.

We agree with this reviewer that this is an overstatement. It was therefore removed.

L. 100: how much did turning on the light affect the chamber temperature? How much of the emissions could be explained by the known exponential relationship between temperature and saturation vapor pressure of the different compounds? The latter could be included in the analysis. Any eventual parameterization of these emissions (say included in a model such as MEGAN) would require these temperature-emission relationships, so this could actually be really useful information that could come from this experiment.

The temperature raised from 26 to 31 C during the experiment with UV lights switched on. The temperature was however constant for the dark experiments with ozone only. We now added figures showing the temperature variations for all three experimental conditions in Appendix - A.

We thank the reviewer for this valuable addition to our paper, we calculated the correlation between the temperature value and the BVOC emissions. We added a new graph to our paper substituting Figure 7 (following reviewer 2 suggestion). In this figure, we selected the 8 most correlated compound with the temperature (Spearman coefficient > 0.8).

We also added the following discussion to these results:

**4.3 Temperature effect on the BVOC emissions**

Higher temperatures increase chemical reaction rates, cellular diffusion rates, and vapor pressure of the VOCs, as a consequence BVOC emission rates are dependent on temperature. In this study, we identified 8 VOC emitted from rapeseed litter which are highly correlated with temperature. Among the most correlated ones, we identified methanol and MEK, in agreement with previous reports investigating such temperature dependence from rapeseed plants (Gonzaga Gomez et al., 2019). Harley et al., (2007) detected methanol emissions from 6 different plant species. Their results reported a correlation between its emission, temperature of the leaves, and stomatal conductance. The mechanisms behind this behavior have been explained by Niinemets and Reichstein, (2003). Methanol is produced within the cell walls, and it diffuses in the liquid phase following the diffusion gradient until it reaches the surface of the cell walls. Then, methanol diffuses in the gas phase into the substomatal cavity and is released as VOC in the ambient air through the stomata. In our study, stomata lock-open as a consequence of cellular death (Prats et al., 2006) and the increased temperature accelerated the diffusion process releasing methanol as the most emitted compound from rapeseed leaf litter.

[Figure]

**Figure 7. Correlation between VOC mixing ratios and temperature under the UV_O3 condition. The 8 most correlated VOC are shown(Pearson correlation coefficients > 0.8).**

Figure 3: is each bar an average of the entire day? Just during light-on conditions? Or an entire 24-hour period? This is unclear. Also, the legend isn't necessary here. Each bar corresponds to the x-axis which already indicates the day. The day does not also need to be indicated with a different color. The different colors could be used to compare different treatments on the same graph (especially if more than one replicate was conducted for each condition), but it doesn't make sense to have the different colored bars in this context.

Each bar represents the entire 24-h period including the dark phase. We redraw this figure following the suggestions of reviewer 2. Please find the new version below with a new caption:

[Figure]

**Figure 3: Summed VOC concentrations for each day (24-h period) incubation condition a) UV, b) O3 and c) UV_O3 Letters indicate the statistical difference obtained by the Tukey test.**

Figure 4: very unclear how the data was organized to conduct the PCA. Some conditions have way more data points than others. It also appears that the authors are using multiple points along the same time-series as independent datapoints for the PCA. This is not appropriate. Are the authors using each individual measurement at each measurement time-point from the PTR for the analysis? Or some smoothed (say 5-minute averaging interval) measurement as an independent data point? A PCA should not be performed with time-series data in this manner. Two datapoints in a single time series are not independent data points in the context of the analysis being conducted here. PCA should be used to compare discrete, independent datasets. Based on the methods, it looks like only one experiment was conducted for each condition and thus, you would only have one multivariate datapoint for each condition (3 total). "multivariate" referring to the entire VOC emission profile. At best, you might be able to argue for using the average emission profile from each day as a single multivariate data point. Ultimately, this needs better clarified, though.

We agree with the reviewer that the PCA needed some clarification. We modified the PCA as suggested by this reviewer by comparing discrete and independent datasets and thus comparing the 6 days for each condition. Each point now represents 1 day of measurement and so even they appear to be superposed for the condition O3 and UV_O3 we compared 6 days measurement for each condition. Please find the below of this revised treatment.:

[Figure]

Figure 4: VOC profiles differences between UV light, UV_O₃, and O₃ conditions, each point represent 1 day of measurement. The percentage of the variance explained by the 2 first components is shown on each axis (Dim1 and Dim2

L. 201: how are you calculating any "statistical difference" with an N=1 representing each condition?

The statistical difference in this case refers to the number of days for each condition, and thus the number of samples is N=6.

L. 209: Authors state "the number of particles decreased" after the initial nucleation. However, the methods state there was a particle wall loss correction applied to the data. Shouldn't this have eliminated the observed decrease in particle number? If not, it seems like the particle wall loss correction was not adequate. How else would they be losing particles?

Conditions in this chamber, and the particle size distribution described, likely wouldn't lead to substantial coagulation, correct?

The particle decrease could be due to coagulation and dilution (due to the flow mode used here). In figure 7, showing SOA formation as a function of particles number concentration, we can indeed see a decrease of the particles number. Below, we report a similar variation using the particle volume, and obviously the same trend is observed. We mainly attribute this variation to a source strength, which is reducing with time, and to the dilution occurring in our chamber. In fact, the chamber was continuously flushed with an air flow to compensate the air withdrawn by the various analysers connected to it.

[Figure]

Section 3.3: I think the authors intend to refer to Figure 7, not Figure 8. It is also very unclear why the analysis was conducted this way. What does a negative correlation between VOC mixing ratio and particle number really tell us? Is that information meaningful? Why conduct this analysis using particle number? It is well established that gas-particle partitioning increases with increasing SOA mass. How much of the differences in partitioning behavior could be explained by increased absorption due to increased mass? The relevance of this analysis is unclear. The correlation doesn't necessarily indicate the compounds that contributed to SOA production. Perhaps they were just the most reactive in the gas-phase. Some modeling approaches could be used here to better understand the chemistry occurring in the chamber. As is, this analysis is very preliminary. More synthesis is required to make this data meaningful.

We changed the Figure 7 by substituting it with the correlation between BVOC emissions and the temperature as reported above. We also agree with this reviewer that the correlation between the SOA and the number of particles does not necessarily mean that the VOC correlated are precursors of the SOA. Therefore, this section was removed.

Section 4.4: How does the mass of SOA generated here (and scaled to an ambient field environment) compare to typical measured PM? It looks like it would be a relatively minor

source of aerosol based on the results shown, but a more convincing comparison could be made using some simple box modeling calculations.

Comparing the measurements made here with the data of a monitoring station is far from being obvious, as we observed new particle formation events (i.e., characterized with low particle mass but high number concentration) and not the mass of PM1 (typically observed in such networks). Therefore, if we simply extrapolate our data to the regional conditions, it will show that these data have little to no significance when considering the particle mass, but a major one when it comes to number concentration. However, such extrapolation would not be scientifically sound, as only a mesoscale modelling approach would allow a proper assessment of the associated regional importance. However, this was clearly beyond the scope of this investigation, submitted as a measurement report.

**Technical Comments**

Too numerous for me to list here. I recommend sending to an editing service.

We thank the reviewer for the comment. We will follow that advice i.e.; a professional proofreading service will correct our English in the revised manuscript.

**References**

Alpert, P.A., Ciuraru, R., Rossignol, S., Passananti, M., Tinel, L., Perrier, S., Dupart, Y., Steimer, S.S., Ammann, M., Donaldson, D.J., George, C., 2017. Fatty Acid Surfactant Photochemistry Results in New Particle Formation. Sci Rep 7, 12693. https://doi.org/10.1038/s41598-017-12601-2

Fischer, R.A., Byerlee, D., Edmeades, G., 2014. Crop yields and global food security: will yield increase continue to feed the world? ACIAR Monograph No. 158. Australian Centre for International Agricultural Research. Canberra.

Gonzaga gomez, L., Loubet, B., Lafouge, F., Ciuraru, R., Buysse, P., Durand, B., Gueudet, J.C., Fanucci, O., Fortineau, A., Zurfluh, O., Decuq, C., Kammer, J., Duprix, P., Bsaibes, S., Truong, F., Gros, V., Boissard, C., 2019. Comparative study of biogenic volatile organic compounds fluxes by wheat, maize and rapeseed with dynamic chambers over a short period in northern France. Atmospheric Environment 214, 16 p. https://doi.org/10.1016/j.atmosenv.2019.116855

Harley, P., Niinemets, Ü., Guenther, A., 2007. Environmental controls over methanol emission from leaves. Biogeosciences Discussions 4. https://doi.org/10.5194/bgd-4-2593-2007

Niinemets, Ü., Loreto, F., Reichstein, M., 2004. Physiological and physicochemical controls on foliar volatile organic compound emissions. Trends in Plant Science 9, 180–186. https://doi.org/10.1016/j.tplants.2004.02.006

Niinemets, Ü., Reichstein, M., 2003. Controls on the emission of plant volatiles through stomata: A sensitivity analysis. Journal of Geophysical Research: Atmospheres 108. https://doi.org/10.1029/2002JD002626

Prats, E., Gay, A.P., Mur, L.A.J., Thomas, B.J., Carver, T.L.W., 2006. Stomatal lock-open, a consequence of epidermal cell death, follows transient suppression of stomatal opening in barley attacked by Blumeria graminis. J Exp Bot 57, 2211–2226. https://doi.org/10.1093/jxb/erj186

Tiefenbacher, A., Sandén, T., Haslmayr, H.-P., Miloczki, J., Wenzel, W., Spiegel, H., 2021. Optimizing Carbon Sequestration in Croplands: A Synthesis. Agronomy 11, 882. https://doi.org/10.3390/agronomy11050882

Reviewer 3

This measurement report describes a study of BVOC emissions and SOA production from leaf litter of rapeseed, an important crop in some countries. The influence of uv light and ozone, both separately and together, was investigated. The topic fits well with the scope of ACP and there are few studies available on this topic. The manuscript is well organized but is difficult to read because it needs a thorough editing for English grammar.

We would like to thank this reviewer for his/her insightful and helpful comments. They helped us to significantly improve our manuscript. We also do apologies for the poor English in the initial submission. The revised version will be thoroughly edited for English grammar. Hereafter, please find our point-to-point answer to the comments raised.

The main issues that should be addressed before publication are:

1. There are a lot of unknown compounds and tentatively identified compounds, as is expected with having only PTR-MS measurements. The study would be improved by including a few measurements of rapeseed litter with complementary techniques, such as GCMS, to identify some of these compounds.

Obviously, the rapeseed litter is seasonal, and our group did not had access to GC-MS instrumentations during the experiments. It would therefore be quite difficult to run this additional analysis. However, we do need to stress that performing VOC analysis only by means of a PTR-MS approach is quite standard in the field of atmospheric sciences. We nevertheless increased the number of tentatively identified compounds using the PTR-Viewer tool.

Does the uv light or ozone change the BVOC emission? Measurements of the emission rates in the absence of uv light and ozone should be reported.

We thank this reviewer for his/her valuable comment. The emission factors of BVOC from litter have been already reported in the literature (Bigg, 2004; Derendorp et al., 2011; Faiola et al., 2014; Gonzaga Gomez et al., 2019; Greenberg et al., 2012; J et al., 2020). The purpose of this article was to point out if there is any contribution of biogenic VOC from litter to SOA formation. For this reason, we focused our experiments on the impact of light and ozone. Below, we show the evolution of the summed VOC concentration in the dark and without ozone. It can be seen that it only changes slightly at levels reduced compared to other conditions. This will be stressed in our revised version.

[Figure]

**Figure 1. summed concentration of the VOC detected in the dark without ozone.**

2. How repeatable are these measurements? Biological systems tend to have a lot of variability. Either replicate experiments should be performed or some evidence should be provided to show that the it is expected that results would be similar if the experiment were repeated.

The number of samples available was limited due to its collection procedure. On the day of the collection, the rapeseed litter used for the measurements was made of leaves at the beginning of senescence process. Replicate experiments are difficult to be performed in this case since other samples will have a different degree of senescence and therefore difficult to compare with the first set of experiments. The evolution of the litter over time is accompanied by a change in the colour of the leaves from green to yellow to brown. This is due to a degradation of the metabolism leading to the death of the cells and the degradation of the chlorophyll. To repeat the experimentation, we would need to renew the litter samples the following year.

Nevertheless, we had obviously to define an experimental plan to address the scientific questions underlying to this work. Such a procedure increases the reproducibility of the starting material for each runs performed here (in total 9 runs). We initially performed a preliminary study (not included in our manuscript) where the BVOC emission and SOA formation from rapeseed litter was investigated in the presence of both UV light and ozone (100 ppb). This preliminary testing showed the potential formation of SOA in the presence of light and ozone (see Figure 2 below). This testing showed some reproducibility (with some inherent variability when working with biological samples). We then decided to perform further experiments under complementary conditions (i.e., $O_3$, UV light, or both), to see the impact of each parameter on the BVOC emission and SOA formation. For each condition, the experiments were repeated 2 times. So, the BVOC data are the average of these replicas. However, due to SMPS failure, only one replica by condition was available.

[Figure]

Figure 2. Temporal evolution of particle number and size distribution, ordinate represents the electrical mobility diameter (nm) and the color scale the particle number concentration. Particle formation for the first day of measurement under UV light irradiation and ozone injection combined.

Some other experiments (not shown here) performed in a small simulation chamber (30 L) were performed on the same litter samples (6 samples of soil + litter). Each sample was placed in an aluminium tray of 52.7 x 32.6 x 8 cm, and they were kept in a chamber at 15°C. In these experiments, only the VOCs were measured every 7 days during a 30-day period. The results are in line with the results shown in this study.

3.  The authors state that emissions from rapeseed leaf litter may have been underestimated (Line 327) but they don't say what the current estimates are. Current estimates should be presented and compared with these results. It is also suggested that SOA formation from leaf litter might be important (Line 332) but there is no indication of how the SOA formation they observed compares with other sources. There should be some comparison with the SOA formation that is currently known or at least predicted in models.

New information about the current literature estimations of the corresponding BVOC emissions will be added, as follows.

Greenberg et al., (2012) detected a methanol flux of 1.3 µg m$^{-2}$ h$^{-1}$ from litter corresponding to 0.4 % of the total emission above the canopy, estimated to be 300 µg m$^{-2}$ h$^{-1}$. In this study, the methanol flux from leaf litter ranged from 4.6 to 28.4 µg m$^{-2}$ h$^{-1}$ depending on the experimental conditions. Hence, our results suggest that the contribution to the total above canopy methanol emissions of the rapeseed litter could range from 2 to 10%.

—

For instance, Greenberg et al., (2012) reported a VOC flux for leaf litter under the canopy of 0.3 μg m$^{-2}$ h$^{-1}$, corresponding to the 0.2% of the total above canopy acetaldehyde emissions, while in this study the emission flux ranged from 1.97 ± 0.01 μg m$^{-2}$ h$^{-1}$ for the UV_O$_3$ condition to 26.7 ± 0.2 μg m$^{-2}$ h$^{-1}$ for the UV condition. The total above canopy acetaldehyde emissions reported by Greenberg et al., (2012) were 200 μg m$^{-2}$ h$^{-1}$. As for methanol, our study suggest a higher contribution to the total above canopy acetaldehyde emissions from leaf litter ranging from 2 to 13 %.

—

**Acetone** (C$_3$H$_6$OH$^+$, 59.049 m/z). This compound was largely emitted from litter under UV irradiation. The average contribution of acetone was 13% under UV light, 1.64 % when influenced by both UV and ozone and 2 % when the litter was exposed to ozone only. Acetone has been reported as one of the most emitted compounds by plants and litter (Gonzaga Gomez et al., 2019; Greenberg et al., 2012). Greenberg et al., (2012) reported an average flux of 0.3 μg m$^{-2}$ h$^{-1}$ between 11:00 and 17:00. In this study, the emissions of acetone were 10 times higher under UV irradiation. Based on (Greenberg et al., 2012), the current estimates of litter contribution to the above canopy acetone emissions is 0.1 %. However, the flux reported in table 2 suggest that the litter contribution to acetone emission, in the absence of ozone, could be as large as 6 %.

For the SOA production estimation, we wrote a full paragraph comparing the SOA detected from our study and the SOA detected in previous works also resumed in Table 3. Please find it here:

**Table 1. Comparison of the SOA formation from leaves litter samples reported in this study and the literature.**

| Sample type | Sampling period | Measured particles range (nm) | Type of chamber | Experimental conditions | Maximum peak of aerosol formation ($\mu m^3 cm^{-3}$) | Total Aerosol volume concentration ($\mu m^3 cm^{-3}$) | Volume Contribution of particles < 20 nm | Ref. |
|---|---|---|---|---|---|---|---|---|
| Mix of: Pinus ponderosa, Pseudotsuga menziesii Pinus monticola, Larix occidentalis litter and soil | May-June 2012 | 20-730 | atmospheric chamber (7.7 $m^3$) | 130 ppb of $O_3$ in dry conditions | 0.97-5.43 | - | - | (Faiola et al., 2014) |
| Mix of: Pinus ponderosa, Pseudotsuga menziesii Pinus monticola, Larix occidentalis litter and soil | May-June 2012 | 20-730 | atmospheric chamber (7.7 $m^3$) | Reproducing raining event 130 ppb of $O_3$ | 0.29-2.55 | - | - | (Faiola et al., 2014) |
| Brassica napus litter | June 2019 | 2.5-79.1 | Multiphase simulation chamber (2$m^3$) | 60-80 ppb of $O_3$ | 0.2 | 15.1 | 38% | This study |
| Brassica napus litter | June 2019 | 2.5-79.1 | Multiphase simulation chamber (2$m^3$) | Only UV light | 0.8 | 85.4 | 24% | This study |
| Brassica napus litter | June 2019 | 2.5-79.1 | Multiphase simulation chamber (2$m^3$) | 60-80 ppb of $O_3$ and UV light | 7.6 | 787.8 | 24% | This study |

4. One of the interesting findings is the relatively high contribution of various organic acids but this is not discussed in the text. The current limited discussion on organic acids should be expanded.

We now added a new paragraph to the discussion to stress this aspect more.

**Acetic acid ($C_2H_4O_2$, 61.03 m/z) and formic acid ($CH_2O_2$ 47.02 m/z).** Organic acids such as acetic and formic acid are mostly emitted from living plants (Kesselmeier and Staudt, 1999), and by the foliage of trees and crops with a flux of 35 µg m$^{-2}$ h$^{-1}$ (Paulot et al., 2011). Viros et al., (2021) detected acetic and formic acid also from senescent litter, with a flux of 0.05 and 0.98 µg m$^{-2}$ hr$^{-1}$, respectively. In this study, the emission rates of the two organic acids ranged from 0.76 to 64.28 µg m$^{-2}$ h$^{-1}$ for acetic acid, and from 0.23 to 9.12 µg m$^{-2}$ h$^{-1}$ for formic acid. Mozaffar et al., (2018) described that the acetic acid emissions were affected by temperature, as they recorded lower emissions in the early morning than during the late afternoon. This could explain the higher emissions of acetic acid observed in our study where the temperature reached 30 °C, higher than the conditions encountered by Viros et al., (2021)  i.e., 22 °C.  Mozaffar et al., (2018) while analysing BVOCs from senescent maize leaf litter, reported an acetic acid contribution to the total BVOC emission of up to 26 %. Similar results were obtained in our study, where under UV light conditions, the contribution of the acetic acid reached 20% of the total VOC emissions.

5. The measurement technique appears reasonable, but the description is lacking and needs more details such as standards, accuracy and precision, etc.

The analytical procedure used during this study is standard, and we add to our main text some additional details as follows.

The PTR-TOF-MS has a mass resolution of 4500 m/Δm. A calibration gas standard (TO-14A Aromatic Mix, Restek Corporation, Bellefonte, USA) containing XX VOCs at a concentration of 100 ± 10 ppb in nitrogen was used to calibrate and regularly assess the instrument performance, including mass resolution, mass accuracy, sensitivity, and relative mass-dependent transmission efficiency. The sensitivity of these compounds ranged between 15 and 70 cps/ppb depending on the actual mass. However, since it was not possible to calculate the exact sensitivity for all the detected compounds, we assumed that the proton reaction constant was always equal to 2 × 10$^{-9}$ cm$^3$ s $^{-1}$ (Cappellin et al., 2011; Kalalian et al., 2020)  and thus the average sensitivity of 30 cps/ppb was applied for all the compounds.

Specific points:

Table 1: It would be useful to report the "per mass" emission in addition to the reported per area emission (or at least provide the specific leaf area so readers can do this calculation) to enable comparisons with literature values.

The emissions were reported here per area, as this is the standard units used for these measurements in the literature. However, we have also provided the mass information for the sample investigated in the caption of table 1 (not to overload the table with information).

Also, this information is available in in the material and method sections:

Once acclimatized, leaves have been weighted and spread on to cover the whole surface of a FEP (fluorinated ethylene propylene) film (with a surface of 0.64 m$^2$) (**Figure 1a**). After 6 days of measurement, the surface covered by the rapeseed litter has been estimated to be 0.45 m$^2$ (**Figure 1b**).

And

The initial weight of rapeseed in the chamber ranged from 75 to 80 g. After 6 days of measurement, the weight decreased by 29-32 %.

Line 25: The authors note that VOCs are either "anthropogenic, related to human activities, or biogenic" and then go on to label emissions from rapeseed as biogenic. But since rapeseed is a crop grown by humans, shouldn't this be considered anthropogenic?

This comment is a valuable one, and should certainly initiate some discussions on how to qualify agriculture. In the present work, we simply stick to the wording commonly used in this subfield of research and qualified VOC emissions from plant, leaves or litter as biogenic. (Mozaffar et al., 2018)

Line 199: Define the Shannon index.

A definition of the Shannon index is now added in the material and method section, please find it here:

Finally, the calculation of the Shannon index was performed. The Shannon index is a quantitative measure reflecting how many different VOC were emitted from each sample. It was calculated with the diversity function of the vegan package (version 2.4-3) in the R software (version 3.2.3). The diversity index was calculated as $H = \sum_{voc} E_{voc} \log(E_{voc})$, where the sum is over all VOCs recorded in the mass table.

Line 292: The authors state that "mature leaves are known to emit less isoprene than young leaves". The referenced papers report the opposite (mature leaves emit more than young leaves) as do other studies. In any case, it should be noted that this isoprene emission from rapeseed leaf litter is not likely to be the same process as

from living plants (whether they are mature or young) but is likely from bacteria or other non-enzymatic production of isoprene.

The paragraph has been rephrased and corrected as follows:

Isoprene (C5H8H+, 69.07 m/z). In this study, isoprene was the 30[th] most emitted compounds only in the experiment without $O_3$. Its average contribution in the UV light experiment was 1% with a flux rate of $3.00\pm0.03$ µg m$^{-2}$ h$^{-1}$ or 0.02 µg g$^{-1}$ h$^{-1}$ which is almost 20 times lower than the emissions reported by Morrison et al., (2016), where the maximum detected flux of isoprene from rapeseed was 0.35 µg g$^{-1}$ h$^{-1}$. This difference is probably due to the different samples, indeed Morrison et al., (2016) investigated branches, while here only the emissions from senescent leaves were considered. However, the flux rate of isoprene reported by this study is in line with those reported by Gonzaga Gomez et al. (2019) i.e., 0.035 µg g$^{-1}$ h$^{-1}$. Isoprene can also be emitted from microorganisms such as bacteria and fungi. Isoprene is an intermediate product of the mevalonate pathway, which lead to the production of essential organic compounds within the microorganisms cells (Hess et al., 2013). Isoprene is therefore a metabolite directly related to the presence of microorganisms in soil and plants (Hess et al., 2013)

Line 309-310. This sentence is confusing, and the meaning is not clear.

This sentence has been removed following the recommendation of reviewer #2.

**References**

Bigg, E.K., 2004. Gas emissions from soil and leaf litter as a source of new particle formation. Atmospheric Research 70, 33–42. https://doi.org/10.1016/j.atmosres.2003.10.003

Cappellin, L., Biasioli, F., Granitto, P.M., Schuhfried, E., Soukoulis, C., Costa, F., Märk, T.D., Gasperi, F., 2011. On data analysis in PTR-TOF-MS: From raw spectra to data mining. Sensors and Actuators B: Chemical 155, 183–190. https://doi.org/10.1016/j.snb.2010.11.044

Derendorp, L., Holzinger, R., Röckmann, T., 2011. UV-induced emissions of C2 - C5 hydrocarbons from leaf litter. Environ. Chem. 8, 602. https://doi.org/10.1071/EN11024

Faiola, C.L., VanderSchelden, G.S., Wen, M., Elloy, F.C., Cobos, D.R., Watts, R.J., Jobson, B.T., VanReken, T.M., 2014. SOA Formation Potential. of Emissions from Soil and Leaf Litter. Environ. Sci. Technol. 48, 938–946. https://doi.org/10.1021/es4040045

Gonzaga gomez, L., Loubet, B., Lafouge, F., Ciuraru, R., Buysse, P., Durand, B., Gueudet, J.C., Fanucci, O., Fortineau, A., Zurfluh, O., Decuq, C., Kammer, J., Duprix, P., Bsaibes, S., Truong, F., Gros, V., Boissard, C., 2019. Comparative study of biogenic volatile organic compounds fluxes by wheat, maize and rapeseed with dynamic chambers over a short period in northern France. Atmospheric Environment 214, 16 p. https://doi.org/10.1016/j.atmosenv.2019.116855

Greenberg, J.P., Asensio, D., Turnipseed, A., Guenther, A.B., Karl, T., Gochis, D., 2012. Contribution of leaf and needle litter to whole ecosystem BVOC fluxes. Atmospheric Environment 59, 302–311. https://doi.org/10.1016/j.atmosenv.2012.04.038

Hess, B.M., Xue, J., Markillie, L.M., Taylor, R.C., Wiley, H.S., Ahring, B.K., Linggi, B., 2013. Coregulation of Terpenoid Pathway Genes and Prediction of Isoprene Production in Bacillus subtilis Using Transcriptomics. PLOS ONE 8, e66104. https://doi.org/10.1371/journal.pone.0066104

J, V., C, F., H, W., J, G., C, L., E, O., 2020. Litter of mediterranean species as a source of volatile organic compounds. Atmospheric Environment 242, 117815. https://doi.org/10.1016/j.atmosenv.2020.117815

Kalalian, C., Letizia, A., Depoorter, A., Lunardelli, B., Perrier, S., George, C., 2020. Influence of indoor chemistry on the emission of mVOCs from Aspergillus niger molds. Science of The Total Environment 741, 140148. https://doi.org/10.1016/j.scitotenv.2020.140148

Kesselmeier, J., Staudt, M., 1999. Biogenic Volatile Organic Compounds (VOC): An Overview on Emission, Physiology and Ecology. Journal of Atmospheric Chemistry 33, 23–88. https://doi.org/10.1023/A:1006127516791

Morrison, E.C., Drewer, J., Heal, M.R., 2016. A comparison of isoprene and monoterpene emission rates from the perennial bioenergy crops short-rotation coppice willow and Miscanthus and the annual arable crops wheat and oilseed rape. GCB Bioenergy 8, 211–225. https://doi.org/10.1111/gcbb.12257

Mozaffar, A., Schoon, N., Bachy, A., Digrado, A., Heinesch, B., Aubinet, M., Fauconnier, M.-L., Delaplace, P., du Jardin, P., Amelynck, C., 2018. Biogenic volatile organic compound emissions from senescent maize leaves and a comparison with other leaf developmental stages. Atmospheric Environment 176, 71–81. https://doi.org/10.1016/j.atmosenv.2017.12.020

Paulot, F., Wunch, D., Crounse, J.D., Toon, G.C., Millet, D.B., DeCarlo, P.F., Vigouroux, C., Deutscher, N.M., González Abad, G., Notholt, J., Warneke, T., Hannigan, J.W., Warneke, C., de Gouw, J.A., Dunlea, E.J., De Mazière, M., Griffith, D.W.T., Bernath, P., Jimenez, J.L., Wennberg, P.O., 2011. Importance of secondary sources in the atmospheric budgets of formic and acetic acids. Atmospheric Chemistry and Physics 11, 1989–2013. https://doi.org/10.5194/acp-11-1989-2011

Viros, J., Santonja, M., Temime-Roussel, B., Wortham, H., Fernandez, C., Ormeno, E., n.d. Volatilome of Aleppo Pine litter over decomposition process. Ecol. Evol. https://doi.org/10.1002/ece3.7533

---

## Author Response (AR2)

Villeurbanne, le 28 juillet 2021

**Jason D. Surratt**
**Atmospheric Chemistry and Physics**
**Editor**
**Special Issue: Simulation chambers as tools in atmospheric research (AMT/ACP/GMD inter-journal SI)**

*Title: Measurement report: Biogenic VOC emissions profiles of Rapeseed leaf litter and their SOA formation potential*
*Author(s): Letizia Abis et al.*
*MS No.: acp-2021-135*
*MS type: Measurement report*

Dear Jason,

We are hereby submitting our revised manuscript entitled "Biogenic VOC emission profiles of rapeseed leaf litter and its SOA formation potential" by Abis et al.

We noted that 2 reviewers out of 3 were recommending the publication of our manuscript while the second reviewer raised again some concern about the repeatability of the starting conditions of our experiments.

As it can be seen in the point-to-point answer, we clearly show that the initial conditions were comparable for all set of experiments. This clearly address the concerns raised by this reviewer. We are therefore confident that an intercomparison between the chosen conditions is meaning full (bearing in mind the obvious variability of biological matter)

I believe that this revised version now meets ACP's criteria as a measurement report, and hope it will be accepted.

Yours sincerely,

Christian GEORGE

[Figure]

**IRCELYON ● UMR 5256**
**2 avenue Albert Einstein ● F-69626 Villeurbanne Cedex**
**tél. +33 [0] 472 431 4 89 ● fax: +33 [0] 472 448 438**
**Christian.George@ircelyon.univ-lyon1.fr ● http://www.ircelyon.univ-lyon1.fr/**

**Editor's comments**

One of the major concerns remaining that this reviewer had is that you have not provided any context for how the emissions are changing over time from this single litter sample completely independent of your ozone and UV treatments. This reviewer strongly indicated that you surely have some "initial" emission data, which you could provide for each treatment trial just to give some sense of how that baseline emission value is changing.

> We are showing below by answering to reviewer 2, that our starting conditions were comparable for all type of experiments performed here due to the storage conditions of the leaf litter sample.

The main remaining issue for this reviewer is regarding the need for replicates. The lack of replicates makes it difficult to make meaningful conclusions about the observations. However, there are several reasons why the paper could be considered suitable for publication anyway.

> Thanks for these comments. We also noted that 2 out of 3 reviewers recommended publication as is. We show below that the experimental conditions were reproducible and for each condition, and despite the limited number of samples each experiment was repeated twice. This is not a perfect situation but it is as good as possible for such type of experiments.

**Reviewer 2**

The revised manuscript has addressed some of the concerns raised previously, but critical flaws remain in data interpretation and presentation. Consequently, I cannot recommend this paper for publication.

General Comments

Any description and discussion of the results needs to make it abundantly clear that differences in VOC emission behavior observed between the different experimental conditions cannot be directly compared because the same leaf litter sample was used throughout, and emissions will change over time as litter decomposes. I understand seasonal constraints on these types of experiments very well, but the interpretation of results needs to be provided within an appropriate context. These 3 different conditions can be presented as 3 entirely separate trials, highlighting that the same litter is being used throughout so the BVOC emissions were changing over time completely independent of the experimental condition imposed. The emissions at the beginning of each condition were likely very different even before the UV, O3, or UV_O3 was applied, so making comparisons about how O3, UV, or UV_O3 influence BVOC emission behavior on leaf litter that was 2 days old at the start of one condition, 9 days old at the start of another condition, and 15 days old at the start of another condition isn't meaningful (I'm guessing on the ages because that information was not provided clearly). The interpretation and discussion of results is still very focused on making comparisons between the different conditions instead of looking at each condition as a

completely separate trial using litter with very different starting characteristics at the beginning of the experiment.

The main concern raised here by this reviewer is that the starting conditions of each of our experiments were different and therefore those cannot be intercompared. This is not correct.

To clarify this, we now added (and corrected) our manuscript with the following information:

*To avoid inhomogeneous samples in terms of the decomposition stage, all of the leaves were cut directly from the stems but making sure that they were falling or about to fall. Overall, 3 kg of leaves were collected from different plants in the field (field area around 1 km2). The rapeseed litter used for the measurements was made of leaves at the beginning of senescence. The leaf samples were stored for at -20 °C. The sampled litter was reused for all the measurements, throughout the experimentation, defrosting just the fraction of sample needed for the experiment. At the beginning of each experiment, the leaves had visually the same aspect and identical mass to volume ratio (as an indirect metric of their decomposition). In addition, the VOCs were monitored during the stabilisation of the experimental conditions and showed identical patterns.*

To demonstrate that intercomparison is indeed possible, we plotted below the total VOCs measured for 2 hours prior the start of each condition tested here (i.e., in the dark, prior to ozone injection, prior to switching on lights in presence of ozone). It can clearly be seen that those starting conditions are in the same range. We therefore assumed that the following evolution is due to the actual experimental plan.

[Figure]

PCA analysis: the authors state that the VOC emission profiles of the different conditions were "strongly different". First, the time/decomposition component cannot be separated out from the experimental condition component. We need to see what the emissions looked like at the start of each trial BEFORE the experimental condition was imposed (essentially a comparison of the "initial emissions" for each condition). Second, PCA does not provide any measure of statistical significance and cannot be used to make any statements about the strength of difference between samples. You can use a PCA to show clustering, but statistical significance has to be measured with a different approach, such as an analysis of variance (ANOVA).

> We agree with the reviewer that that no statistical differences can be highlighted with the PCA. However, the PCA can be used to visualize possible differences between VOC emissions profiles. For this reason, we changed the sentence as follow:
>
> > "The PCA shows that the VOC profiles emitted during the UV condition were separated from the VOC profiles emitted from the UV_$O_3$ and $O_3$ conditions"
>
> We clarified, with the text above, the comparison of the initial emissions.

Figures 4, 5 and 7 are not publication quality at the standard generally seen in ACP.

> We now improved the quality of these figures by increasing the resolutions for figure and changing the layout of figure 7. For the PCA plot (figure 4), we believe that this standard presentation is adequate, but we nevertheless increased slightly its quality.

Now that it is clear the same leaf sample was used for each condition, there are other ambiguities in the methods section that need to be clarified. In Section 2.1 on "sample collection", how many leaves/branches were collected? From how many different plants? It says a "random sampling method" was used, which made more sense before it was clear that the same set of leaves was used throughout the experiment. What does "random sampling method" mean in this context? Were a certain number of leaves collected from a random sampling of plants? Or were X number of leaves randomly sampled from the same plant? This is still unclear.

> To clarify this, we now added (and corrected) our manuscript with the following information (identical to above):
>
> > *To avoid inhomogeneous samples in terms of the decomposition stage, all of the leaves were cut directly from the stems but making sure that they were falling or about to fall. Overall, 3 kg of leaves were collected from different plants in the field (field area around 1 km2). The rapeseed litter used for the measurements was made of leaves at the beginning of senescence. The leaf samples were stored for 2 days at -20 °C for the first sample used, 7 days for the second sample used and 15 days for the third sample before measurement. The sampled litter was reused for all the measurements, throughout the experimentation, defrosting just the fraction of sample needed for the experiment. At the beginning of each experiment,*

*the leaves had visually the same aspect and identical mass to volume ratio (as an indirect metric of their decomposition). In addition, the VOCs were monitored during the stabilisation of the experimental conditions and showed identical patterns.*

Now that it is clear the same sample was used for each condition, Table 1 should indicate the dates for each condition. The leaves would have been decomposing over time (and their emissions changing as a result). It would be useful to have some sense for the "age" of the leaf litter during each condition.

As the initial conditions were comparable and no decomposition occurring due to the storage conditions, we do not believe that adding the dates of experiments is a useful information.

Section 2.4 Experimental Set-up should make is explicitly clear that the same leaf litter is being used for all three experimental conditions studied. In my opinion, there is still some ambiguity about that as written.

We now believe that the text above clearly answers also to this comment.